# A hybrid dense convolutional network and fuzzy inference system for pneumonia diagnosis with dynamic symptom tracking

Sulav Baral[1‡], Rabindra Bista[1‡], Sanjog Sigdel[1], João C. Ferreira[2,3]*

**1** Department of Computer Science and Engineering, School of Engineering, Kathmandu University, Dhulikhel, Kavre, Nepal, **2** Faculty of Logistics, Molde University College, Molde, Norway, **3** INOV INESC, Lisbon, Portugal

‡ First Author.
* joam@himolde.no

## Abstract

### Background

Pneumonia is a major cause of mortality among children under five and adults over 65, especially in low-resource settings where access to skilled radiologists is limited. Accurate and early diagnosis is essential, but is often hindered by subjective interpretation and variability in its symptoms.

### Objectives

This study aims to develop a hybrid Artificial Intelligence (AI) based pneumonia diagnosis system that integrates Deep Learning (DL) confidence scores, DenseNet201 with Capsule Network (CapsNet), Mamdani-style fuzzy inference, and a dynamic symptom adjustment mechanism to enhance diagnostic accuracy, transparency, and clinical usability.

### Methods

The system was evaluated using 17,229 labelled chest X-ray images across multiple cross-validation techniques: Stratified, k-fold, Bootstrap, and Monte Carlo methods, each with five dataset iterations or folds. DenseNet was used to extract spatial features, while CapsNet preserved spatial orientation and hierarchical relationships. A DL based confidence score was generated and used as a fuzzy membership input to support classification in borderline cases, where severity scores were nearly tied, and the confidence score guided the final decision. A dynamic adjustment algorithm further refined symptom severity by incorporating recent trends in patient data.

**Data availability statement:** All data underlying the findings of this study are publicly available and can be accessed through the Kaggle platform. The original dataset, titled "Chest X-Ray Images (Pneumonia)", is maintained by the original uploader and can be found at the following URL: https://www.kaggle.com/datasets/moazeldsokyx/chest-x-ray-images-pneumonia-dataset. Furthermore, the trained ML models and the codes implemented for this research can be found at public GitHub repository through https://github.com/sulav1995/Pneumonia-Diagnosis-with-ML-and-Fuzzy.

**Funding:** The author(s) received no specific funding for this work.

**Competing interests:** The authors declare no conflict of interest.

## Results

The DenseNet201 + CapsNet architecture achieved the highest performance in the 5th fold of stratified cross-validation, with a test accuracy of 99.01%. The model also demonstrated strong generalization, with a weighted precision, recall and F1-score of 0.9878, 0.9874, and 0.9876, respectively, across all classes. The paired t-test confirmed that the CapsNet-based approach outperformed traditional fully connected layers, and the fuzzy logic system effectively handled ambiguous cases using DL confidence. The dynamic membership mechanism showed strong adaptability for real-time symptom tracking.

## Conclusion

This hybrid model offers a robust, interpretable, and clinically relevant decision-support tool for pneumonia diagnosis. It bridges high-performance AI with real-world medical decision-making, especially in settings with limited radiological expertise.

## 1. Introduction

Pneumonia continues to pose a global threat [1,2], particularly among children under 5 [3], and adults over 65. Common clinical symptoms that patients may encounter include fever, coughing, confusion, Dyspepsia and chest discomfort. Early diagnosis of pneumonia is critical to ensure proper treatment and increase survival rates, particularly in vulnerable populations such as Children and the elderly [4]. However, overlapping symptoms with other respiratory illnesses and substantial inter-observer variability among radiologists sometimes make it difficult to manually interpret Chest X-rays (CXR), the most widely used diagnostic technique for pneumonia. AI-powered tools offer high accuracy, but most are static and struggle to adapt to the changing symptom severity over time. Additionally, they frequently serve as "black boxes," providing little interpretability for clinical judgments made in real time [5].

In recent years, CNN (Convolution Neural Network)-based architectures have been demonstrated to exhibit excellent performance in medical image classification, including CXR-based pneumonia diagnosis. CNNs automatically learn spatial hierarchies of features from images without needing handcrafted features and providing high diagnostic accuracy. Scalability, fast image processing, and strong performance in pattern recognition are the advantages of CNNs. CNNs are prone to being black boxes and giving little transparency into how predictions are made. Moreover, CNN models are static models—they are trained once and do not account for symptom progression or patient condition variability over time [6]. Fuzzy Logic Systems, on the other hand, offer a rule-based decision-making system that mimics human reasoning. They are especially well-suited to handle uncertainty, vagueness, and partial truths and are thus well-suited to interpret symptoms of differing severity. Fuzzy logic provides interpretable outputs and accommodates the integration of domain knowledge in the form of fuzzy rules. However, traditional fuzzy systems are limited by their reliance on predefined rules and do not leverage the pattern recognition ability of deep learning.

Few existing models have tried to combine CNN predictions and fuzzy-based symptom interpretation, but not in a dynamic or adaptive integration of the CNN's confidence scores for severity estimation. This prevents doctors from depending on and using the system for treatment. This work investigates the possibility of precise and comprehensible pneumonia severity categorization using a hybrid deep learning and fuzzy logic system in conjunction with DL based confidence score (also referred as CNN based confidence score) [7] in situations when symptoms are rapidly changing, by introducing a dynamic fuzzy inference engine. The system incorporates CNN confidence scores directly into the fuzzy logic module, improving interpretability and severity scoring in borderline cases—a situation where the fuzzy model, whether it includes CNN confidence or not, struggles to tell apart two severity levels. This situation occurs when the top two severity scores are nearly identical. This work aims to bridge the gap between AI predictions and clinically meaningful assessments, particularly in scenarios where pneumonia symptoms/conditions change over time [8].

The primary contributions of this work are:

- **AI-Driven Pneumonia Diagnosis:** Our hybrid model integrates DenseNet + CapsNet architecture and Mamdani-style fuzzy logic for real-time pneumonia-based symptom/condition tracking, improving interpretability and diagnostic accuracy.

- **Strong generalization with transfer learning:** Using DenseNet201 + CapsNet on 17,229 CXRs, our model demonstrates strong generalization and consistent performance across multiple cross-validation strategies, achieving 99.01% test accuracy. Furthermore, the efficiency of CapsNet has also been compared to a standard FC (Full Connection) based pipeline, via a paired t-test.

- **Integration of DL/CNN based confidence score as a fuzzy membership:** To handle borderline cases, our mamdani style fuzzy logic combines the DL based confidence score (also referred as CNN based confidence score), which was modelled with the help of a clinician.

- **Integration of a dynamic membership adjustment algorithm:** The proposed algorithm adjusts the pneumonia related symptoms/conditions depending upon the whole severity of any patient's condition (the associated conditions). The algorithm also proved to remain stabilized, and prevents sudden fluctuation among the defined membership boundary of each associated pneumonia symptoms/conditions.

The rest of the paper has been organized as follows. In Section 2, we present a literature review. Section 3 talks about the methodology of research. Before concluding in section 5, we discuss the results of our work in Section 4.

## 2. Related work

### 2.1 Chest X-ray imaging and pneumonia disease

The main imaging method for detecting pneumonia is a chest X-ray (CXR) [9], which is accessible and reasonably priced. However, because CXR pictures have characteristics with other lung disorders, it can be difficult to interpret them, and radiologists may diagnose different conditions [10]. Aforementioned, the necessity of automated diagnostic systems has consequently drawn a lot of attention. Artificial intelligence (AI)-driven methods and computer-aided detection (CAD) systems have been created to increase accuracy and decrease CXR-based pneumonia diagnosis diagnostic inconsistencies.

### 2.2 Machine learning and AI in pneumonia diagnosis

Artificial intelligence and machine learning have revolutionized pneumonia identification by automating chest X-ray processing and minimizing human interpretation [11–13]. Deep Learning (DL), particularly convolutional neural networks, has improved pneumonia categorization by extracting complex patterns from massive datasets. AI-driven diagnostic solutions reduce human error and expedite procedures, benefiting radiologists and improving the overall quality of care [14].

## 2.3 CNN models for CXR image processing and classification

Convolution Neural Network (CNN) [15–17] has been widely preferred for Chest X-ray (CXR) to perform image feature extraction. Varshni et al. [18] proposed a work that used CNN's DenseNet-169 model for feature extraction and SVG as a classifier, achieving an AUC of 0.8002. Similarly, the work of Rahman et al. [19] utilized four pre-trained CNN models (AlexNet, ResNet18, DenseNet201, SqueezeNet) to classify 5247 chest X-ray images of normal, bacterial pneumonia, and viral pneumonia proved DenseNet201 being the most effective CNN model for CXR for achieving 98% accuracy. Chutia et al. [20] utilized DenseNet201 for COVID-19 and Pneumonia detection which achieved an accuracy of 95.34%. Sanghvi et al. [21] achieved 99.1% accuracy for detection of COVID and Pneumonia by utilizing DenseNet201 for CXR images. Table 1 discusses the previous works in DenseNet architecture using Chest X-ray images and the results achieved.

DenseNet architectures have shown remarkable metrics [26,27], particularly for extracting image features. However, Deep learning isn't standalone enough due to variability in Pneumonia's symptom pattern, as Xin KZ et al [28] found that deep convolutional neural networks performed well on internal datasets with an AUC of 0.95, but their accuracy fell to 0.54 on external datasets, indicating limited generalizability in different clinical settings.

## 2.4 Fuzzy logic for symptom-based pneumonia diagnosis

Fuzzy logic [29,30] has been utilized to handle imprecise information by combining it with Neural Networks [31]. Fuzzy logic systems are good at managing uncertainty and gradual changes. They have also been used for handling clinical parameters or symptoms associated with pneumonia [32,33] with remarkable accuracy. For instance, the work of Arani et al. [34] created a system to diagnose pneumonia using Mamdani-style fuzzy logic. Their system uses 17 input variables, like body temperature and cough severity, to assess the likelihood of pneumonia, but only relies on rule-based fuzzy inference. They achieved 97% sensitivity, 85% specificity, and 93% accuracy. Regardless of the high-end metrics, static fuzzy rules cannot easily adapt to diverse patient presentations or evolving clinical knowledge. Moreover, these systems may overlook spatial abnormalities visible on radiographs or subtle symptom progression over time without integration with image-based features or temporal symptom tracking.

## 2.5 Confidence-driven diagnosis using DL-fuzzy integration

Deep learning-based confidence scores have improved clinical decision-making by controlling diagnostic uncertainty. Recent studies, such as the "TPat framework" by Tuncer et al. [35] have shown how structured feature transitions can

**Table 1. Previous studies utilizing dense architecture for disease diagnosis.**

| Study | Preferred Model | Target Disease | Accuracy | Remarks |
|---|---|---|---|---|
| Kundu et al. [22] | DenseNet-121 | Pneumonia | 98.81% | Achieved high accuracy; used data augmentation to improve model generalization. |
| Jaiswal et al. [23] | DenseNet-201 | COVID-19 | 96.25% | Achieved highest accuracy on DenseNet201 |
| Rochmawanti & Utaminingrum [24] | DenseNet-121 | Tuberculosis, Pneumonia, Cardiomegaly, COVID-19 | Tuberculosis: 89.2%, Pneumonia: 90.4%, Cardiomegaly: 89.8%, COVID-19: 98.6% | Best accuracy achieved with 224x224 image resolution; Global Average Pooling and dropout used to mitigate overfitting; batch normalization accelerates training |
| Anakha et al. [25] | DenseNet-201 | COVID-19 | 96.54% | DenseNet201 outperformed other models with the best accuracy for COVID-19 detection from X-rays |
| Chutia et al. [20] | DenseNet-201 | Lung diseases: Pneumothorax, and Atelectasis | 95.34% | DenseNet201 proved efficient compared to other models |

enhance transparency and clinician trust in biomedical diagnosis (though intended for Parkinson's detection using FNIRS signals), which offers valuable insight for medical imaging AI. By connecting model features with time-related or clinical changes, TPat offers a path that aligns with our goal of combining DL confidence score with fuzzy logic, and also with our aim to create a more understandable approach to pneumonia classification.

Similarly, a hybrid diagnostic framework for dental pulpitis was developed by Chauhan et al. [36], using fuzzy logic and convolutional neural networks. The system classifies X-ray images into deep or shallow cavities, generates confidence scores, and converts these into fuzzy terms. The fuzzy system, guided by 665 expert-defined rules, produces a diagnosis with a 94% accuracy rate, outperforming expert predictions by 7%. Interestingly, the work of Manna et al. [37] created an ensemble framework based on fuzzy ranks for the classification of cervical cytology. Their approach combines the decision scores of three CNNs trained on the ImageNet dataset: Inception v3, Xception, and DenseNet-169. Using two non-linear functions (tanh and exponential decay) on the SoftMax outputs of each model, they implemented a mathematically guided fusion strategy in place of conventional averaging or majority voting. While the other function measured "deviation" from certainty, the first function quantified the "reward" for predictions near 1. Each class in each model had a fuzzy rank score created by multiplying these two values, and the class with the lowest overall fuzzy rank across all classifiers was chosen to make the final prediction. In both of these studies, we found significant limitations regarding how such DL based probability-based confidence scores are generated. Table 2 highlights the summary of previous studies utilizing DL-based confidence scores.

These research studies have attempted to combine CNN outputs with fuzzy logic for pneumonia diagnosis, but their methods and robustness vary significantly, and their limitations suggest that more visible, calibrated, and interpretable fusion procedures are needed for trustworthy decision-making in clinical AI systems.

## 2.6 Dynamic fuzzy membership adjustment

Lekkas et al. [39] introduced one of the earliest evolving fuzzy diagnosis systems, which actively adjusts fuzzy membership functions in real time using new patient data. Their eClass system uses a fuzzy rule base to evolve by processing each data sample sequentially, allowing the fuzzy sets to change dynamically. The Gaussian membership function is recursively updated to reflect the distribution of data points around each rule prototype, and a buffering strategy improves accuracy and consistency during online learning. However, their method does not incorporate symptom-specific clinical thresholds, percentile-based tuning, or smoothing over time windows. Membership changes are driven by proximity and

**Table 2. Existing literature using DL confidence score and its limitations.**

| Study | Confidence score generation approach | Merit | Limitations |
|---|---|---|---|
| Manna et al. | Applied two non-linear functions (tanh and exponential) to raw SoftMax outputs from three CNNs to compute fuzzy ranks | Introduced a mathematically structured approach to fuse multiple CNN outputs | Relying on uncalibrated SoftMax probabilities can lead to some pretty skewed results. The way those combined non-linear transformations work might either boost or dampen certain confidence values way out of proportion. This can result in distorted risk interpretations and make the fuzzy output harder to understand. |
| Chauhan et al. | Used raw CNN probabilities for deep and shallow cavities as direct fuzzy inputs, mapped into probability ranges | Demonstrated a practical integration of CNN outputs with fuzzy logic for medical imaging | Took the SoftMax scores at face value, without any normalization or calibration. This approach can lead to scores that don't accurately reflect true confidence levels, which might skew severity classifications, particularly when there's a lot of uncertainty in the model. |
| Rakshitha et al. [38] | Combined CNN-based chest X-ray analysis (via VGG16 and ResNet50V2) with symptom-based fuzzy logic to derive pneumonia likelihood | Presented a hybrid framework that combines clinical symptoms/factors with image-based CNN analysis for improved diagnosis accuracy | Their integration approach lacks detail on how CNN outputs are transformed before entering the fuzzy system, leaving uncertainty around the utilization of the CNN-derived confidence values. |

variance relative to previous samples. Ahmed et al. [40] developed a dynamic fuzzy rule-based system for early COVID-19 diagnosis, improving on Lekkas' work. Their system used data distribution techniques to form trapezoidal membership functions from actual survey data, allowing for the realistic simulation of symptom severity based on patient metrics like fever and age. Still, it has limitations, including a lack of personal changes, rigidity in adapting to changing symptoms over time, and reliance on aggregate statistics, which hamper effective real-time or longitudinal diagnostics. Shoaip et al. [8] also introduced a fuzzy diagnosis approach that maps symptom descriptions to severity levels using medical ontologies. The system offers flexibility in clinical interpretation but maintains constant fuzzy membership functions, limiting its application in real-time diagnosis settings due to prioritizing semantic input mapping over fuzzy logic enhancement.

The highlighted limitations in previous studies show the need for a new approach that not only updates fuzzy membership functions but also tracks how symptoms change over time. Instead of using fixed or group-based values, the improved method should include smoothing, adjust for changes in data patterns, and use percentile thresholds based on each patient's condition. It should also allow regular updates during follow-up visits so that the fuzzy logic reflects whether symptoms are getting better or worse, making the diagnosis clearer and more useful in real-time and long-term care.

### 2.7 Advantage over previous works

Our technique utilizes pneumonia classification using a DenseNet201 model and a pneumonia diagnostic pipeline that constructs calibrated confidence scores. Rather than treating SoftMax outputs as final, we employ a multi-component calibration approach that uses entropy normalization alongside class probability margins—also known as opposing class strength—and top-1 class prediction dominance. This improves the confidence score accuracy concerning certainty in diagnosis. Such scores are appropriate for use in a fuzzy inference system and improve severity classification's interpretative and clinical value. Moreover, the fuzzy membership values are continually updated depending on the observed symptom trends and follow-up data, using a variance-aware smoothing algorithm that incorporates recent symptom history, Z-score filtering, and percentile-based thresholds to define an adjusted range, allowing the system to adapt to individual patient-defined progression in real time with explainable and transparent patient outcomes. This is further enhanced by including a Capsule Network that preserves spatial hierarchies, enabling better confidence calibration and integration into the fuzzy logic layer for dynamic severity estimation.

### 3. Methodology

The proposed tool utilizes deep learning and fuzzy logic to evaluate the severity of pneumonia based on chest X-ray images.

As per Fig 1, initially the input X-ray is pre-processed, followed by classification into three categories: Normal, Abnormal, and Pneumonia using a deep learning model. Each of these categories yields a confidence score, which is extracted and supplied to the fuzzy inference system, which then classifies the severity of pneumonia as either negligible, mild, moderate, or severe. We focus on advanced interpretability and adaptability features as the system recalibrates fuzzy membership limits utilizing follow-up symptom data. This enables gradual deep personalization and long-term tracking of severity changes over time.

### 3.1 Data collection

For this study, we used a publicly available dataset consisting of 17,229 CXR image datasets from Kaggle [41]. The dataset is commonly used in academic and clinical research for benchmarking AI-based pneumonia detection systems. It consists of train and test folders, each having three classes of images: "Abnormal", "Normal" and "Pneumonia". As evident from Fig 2, this dataset consists of the standard anatomy of the chest X-ray for Pneumonia Diagnosis, taken in different positions, including both males and females of different ages, though not specified in the original data repository. The resolution varies from a minimum of 232 by 232 pixels to a maximum of 1674 by 1493 pixels.

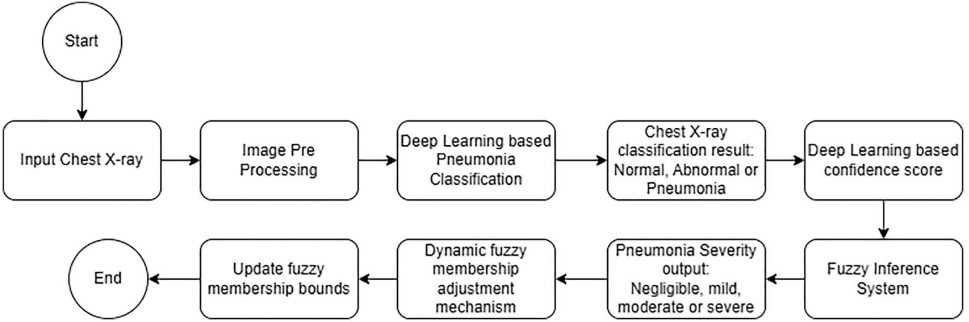

**Fig 1. Flow diagram of hybrid deep learning and fuzzy logic-based pipeline for pneumonia classification and severity assessment with dynamic symptom adaptation.**

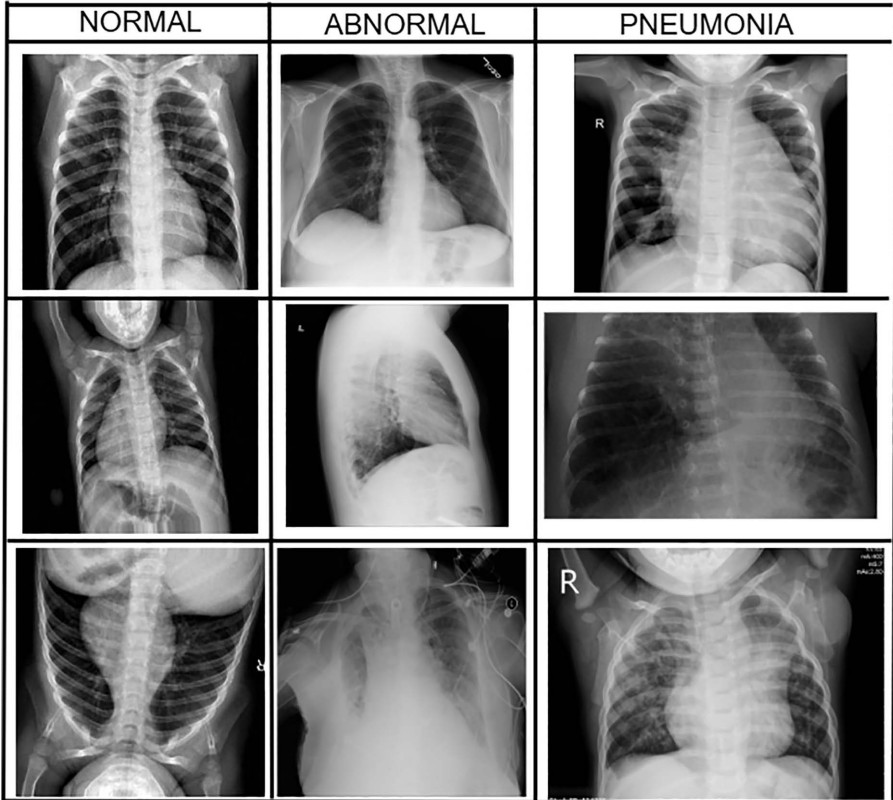

**Fig 2. Chest X-ray images corresponding to "Normal", "Abnormal" and "Pneumonia" classes.**

In the dataset, radiologists assigned labels to the images, and these labels were checked for agreement among different institutions. The label "Normal" refers to healthy individuals. The "Pneumonia" category includes cases where pneumonia has been clinically confirmed, mostly caused by bacteria or viruses. Meanwhile, "Abnormal" is used for chest X-rays that show other issues not related to pneumonia, along with some uncertain findings. These labels come from official radiology reports or hospital systems. While there were steps taken to ensure the labels were accurate, it's important

to note that interpreting radiology without microbiological tests can lead to potential errors in the labels. Overall, this process aims to maintain a clear understanding, even though there are some risks involved.

The dataset is useful for training deep learning algorithms, but it has a few limitations. There are potential prejudices due to pictures from specific hospitals, as well as inconsistent labeling, and missing crucial clinical details like patient history and vital signs. These drawbacks could affect the diagnosis in real-world situations, as they may not apply to global populations or accurately diagnose various health conditions in actual hospital situations.

### 3.2 Splitting the dataset using cross-validation methods

Several strategies were followed in data splitting to reliably test, validate, and train the model. Stratified k-fold cross-validation was used to ensure that each fold had a balanced distribution of data. Monte Carlo cross-validation repeatedly splits the data into random training and validation sets, offering multiple perspectives on model performance. Bootstrap sampling created several training sets through resampling, which checked the model's stability on unseen data. Additionally, k-fold cross-validation was applied to divide the data into equal parts, providing an alternative view of model performance without focusing on class balance.

Depending upon the cross-validation method, the dataset was divided into 5 folds/iterations of Training, Validation, and Test sets using Python scripts (Fig 3). For a consistent comparison of performance, the test set stayed the same. Based on particular reasoning, the remaining data was separated into Training and Validation subsets for each split approach (Stratified k-fold [42], k-fold [43], Monte Carlo [44], and Bootstrap [45]). Bootstrap used samples for training and validation, Monte Carlo randomly divided data into training and validation sets many times, and stratified k-Fold and k-Fold partitioned data into numerous folds for validation. The advantages of these methods are:

- Stratified k-fold cross-validation ensures class distribution maintains the original dataset, crucial in medical imaging where imbalanced classes can negatively impact model performance, such as fewer pneumonia cases.

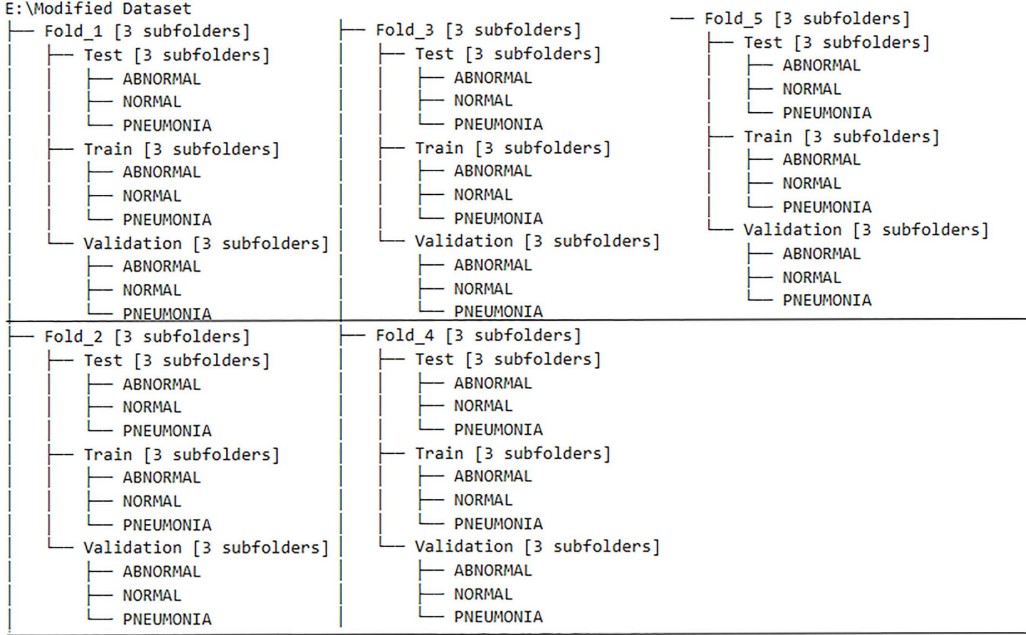

**Fig 3. (Illustration) Directory structure showing train, validation, and fixed test splits across five folds/iterations, organized by class (ABNORMAL, NORMAL, PNEUMONIA) for cross-validation experiments.**

- k-fold cross-validation divides the dataset into k equally sized folds, where the model trains on k-1 folds and validates on the remaining one. This process repeats k times, with each fold serving as the validation set once. It gives a more generalised estimate of the model's performance by rotating through all subsets.

- Bootstrap Resampling generates multiple training sets by randomly sampling the original dataset with replacement, estimating performance metrics like accuracy and AUC due to data distribution shifts.

- Monte Carlo Cross-Validation randomly splits the dataset into training and validation sets multiple times. This method helps detect overfitting and evaluate how well the model holds up under different random train-validation splits.

This methodical division made sure that models were thoroughly assessed across various data configurations. The Comparative performance of cross-validation can be found in the results section of this paper.

Table 3 shows that all cross-validation methods maintain a dataset size of 17,229 images, but the distribution of training, testing, and validation sets varies depending on the approach. Stratified k-fold, Monte Carlo, and regular k-Fold use a consistent split, allocating 12,000 images for training, 3,000 for validation, and a fixed test set of 2,229 images in every fold. Bootstrap method, however, exhibits variation in training and validation set sizes across its five iterations, which draws training data with replacement and uses out-of-bag samples as validation sets. Despite these differences, the fixed test set ensures a standardized evaluation metric for model performance.

### 3.3 Machine learning approach

To diagnose pneumonia, we suggest an integrated approach that combines a fuzzy inference system to classify pneumonia severity levels with a feature extraction process based on DenseNet201, as illustrated in Fig 4.

**3.3.1 Image preprocessing.** The preprocessing phase enhances input images for feature extraction. This phase ensures uniformity and improves the quality of images, addressing challenges related to varying imaging conditions, noise, and limited datasets. The following techniques are applied.

**Contrast Limited Adaptive Histogram Equalization (CLAHE).** CLAHE was applied to improve CXR image quality, especially in low-contrast areas of the CXR. We utilized a clip limit of 2.0, and the image was split into an 8 by 8 tile. The latter

The clip limit was calculated as:

$$H'(i) = min(H(i), clipLimit) \tag{1}$$

After calculating the Clip limit via equation 1, the mapping function calculates the new intensity for each pixel in the tile, enhancing local contrast as:

$$Output(i) = Max\ intensity \ \times \ \frac{Cumulative\ sum\ of\ H'(i)}{Total\ Pixels\ in\ Tile} \tag{2}$$

**Table 3. Dataset split for cross-validation methods among 5 iterations/folds.**

| Cross-validation method | Train | Test | Validation | Total |
|---|---|---|---|---|
| Stratified k-fold | 12000 | 2229 | 3000 | 17229 |
| Monte Carlo | 12000 | 2229 | 3000 | 17229 |
| Bootstrap | 8203, 8245, 8288, 8313, 8266 | 2229 | 6797, 6755, 6712, 6687, 6734 | 17229 |
| k-fold | 12000 | 2229 | 3000 | 17229 |

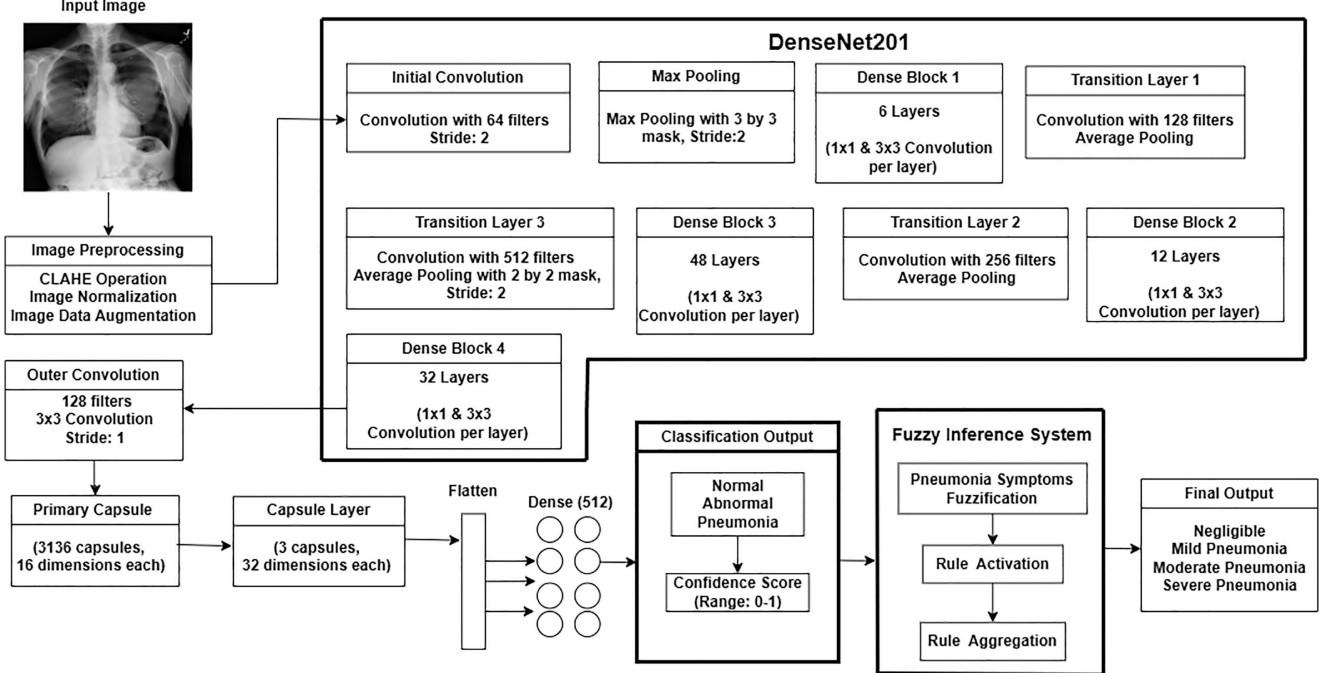

**Fig 4. Flowchart of CNN-based fuzzy inference system for pneumonia severity classification and symptom analysis.**

Once each tile has been processed by equation 2, the tiles are blended to form the final equalized image. To justify the choice of our parameters (clip limit = 2.0 and tile size = 8 by 8), we used 500 randomly selected training images from each class (ABNORMAL, NORMAL, PNEUMONIA) in a sensitivity analysis across various clip limits (1.0, 2.0, 3.0) and tile grid sizes (4 × 4, 8 × 8, 16 × 16) to support our selection of CLAHE parameters. We calculated the mean, standard deviation, and Shannon entropy of the CLAHE-enhanced images for each combination of parameters to measure texture richness and contrast, respectively, which is shown by Fig 5 below.

**Image normalization.** After CLAHE operation, the image is resized to a resolution of 224 by 224. Then, Normalization adjusts the pixel values in images to a similar range, typically 0–1. For an 8-bit image whose maximum pixel value is 255, the normalization is performed as:

$$Normalized\ Pixel = \frac{Pixel\ Value}{255.0}$$

(3)

**Data augmentation.** To simulate real-world imaging variances, especially for chest X-rays, TensorFlow-based data augmentation was utilized to increase model resilience. Spatial diversity was created by the use of techniques such as zooming, shearing, random rotation, and horizontal and vertical shifts. Better pneumonia diagnostic accuracy is achieved by flipping to account for patient orientations, adjusting brightness to simulate changing lighting conditions, and using nearest-neighbor filling to retain anatomical features.

**3.3.2 Image feature extraction.** Though this work also experimented on DenseNet121 and DenseNet169, the DenseNet201 architecture has been leveraged to extract robust features from the preprocessed images, and the reason is provided in the Experiment section of this report. Key components of this architecture include:

- Initial convolution and max pooling: The input image undergoes convolution with 64 filters and max pooling to reduce spatial dimensions.

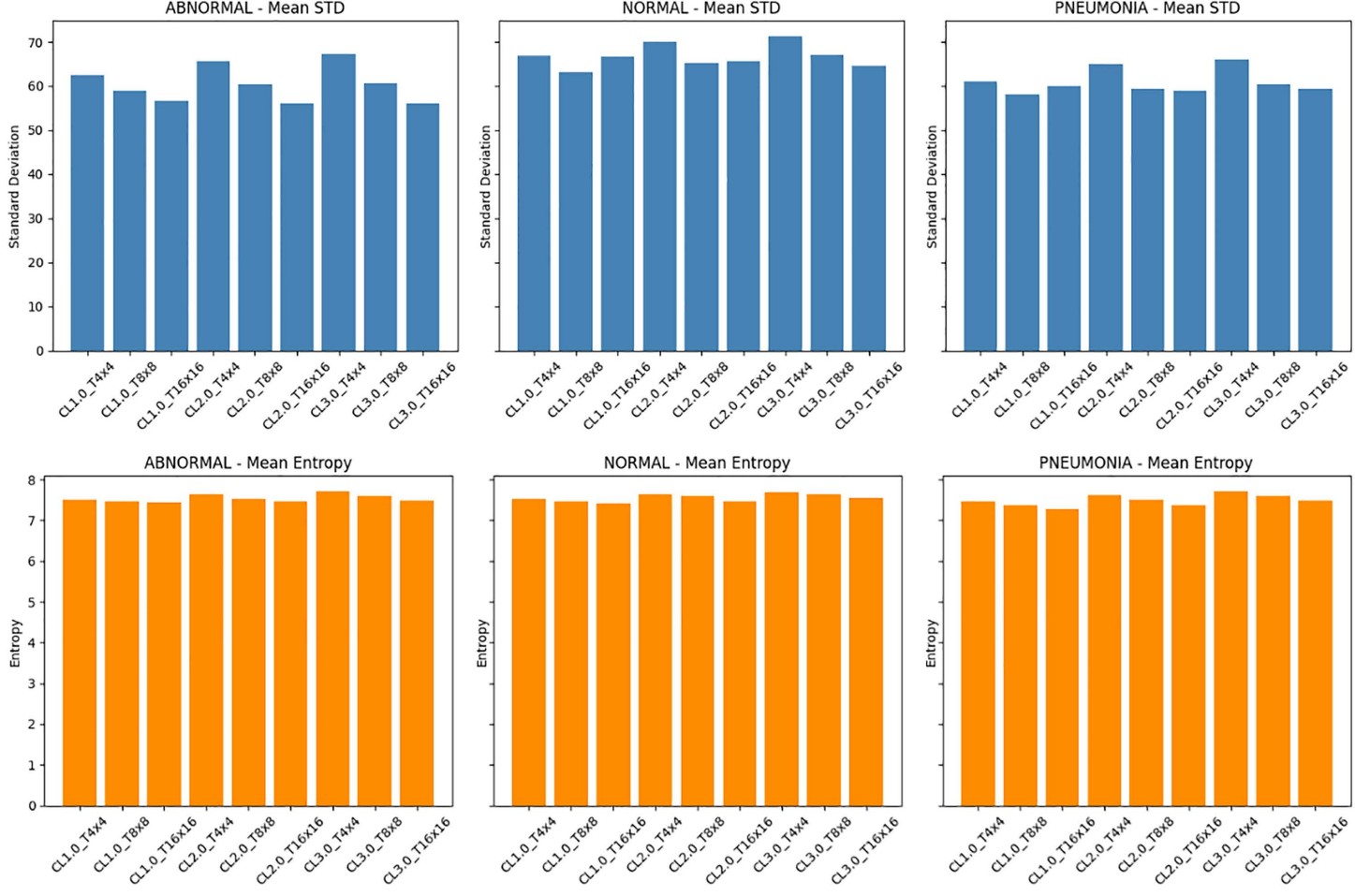

**Fig 5. CLAHE sensitivity analysis on 500 images per class.** Top: mean standard deviation (contrast); Bottom: mean entropy (texture). Clip limit of 2.0 and Tile size of 8x8 achieved the best overall balance. The ensuing bar plots demonstrate that a clip limit of 2.0 consistently resulted in higher entropy and somewhat higher contrast for every class. Both the 4 × 4 and 16 × 16 tile sizes showed good performance, but the 8 × 8 arrangement provided a balanced improvement in all situations without causing undue local amplification.

- Dense blocks and transition Layers: DenseNet201 employs four dense blocks interconnected by transition layers. These layers use batch normalization, ReLU activation, and convolutional filters for feature extraction. Outputs are progressively refined, culminating in high-dimensional feature maps.

- Feature bottleneck: The final dense block produces a feature tensor, which undergoes further processing through custom convolutional layers and capsule layers.

### 3.3.3 Capsule network for classification.

The feature tensor is passed through a capsule network [46], enabling dynamic routing and hierarchical feature representation

**Primary capsule layer: vectorization of feature maps.** These transform feature maps into capsule vectors, preserving spatial hierarchies. We first extract deep feature representations from the preprocessed chest X-ray image using a convolutional backbone (Feature bottleneck layer). This feature tensor, denoted by $F \in R^{H \times W \times C}$, is generated by passing the image through a DenseNet-based network, followed by an additional convolutional layer to refine feature channels.

We then transform this tensor into a set of low-dimensional capsule vectors using a primary capsule layer. The layer performs convolutional operations to reshape the feature tensor into $N$ capsule vectors of dimension $d$.

$$U = reshape\,(Conv2D(F)) \in \mathbb{R}^{N \times d}$$
(4)

Each capsule $u_i \in \mathbb{R}^d$ is then normalized using the squashing function to ensure that the vector length remains between the ranges of 0–1.

$$v_i = \frac{\|u_i\|^2}{1 + \|u_i\|^2} \cdot \frac{u_i}{\|u_i\| + \in}$$
(5)

The output vector $V = [v_1, v_2, \ldots, v_N] \in \mathbb{R}^{N \times d}$ is the input to the class capsule layer.

**Class capsule layer: dynamic routing and aggregation.** We then route the primary capsules $V$ to a set of class-specific capsules $V' \in \mathbb{R}^{K \times d'}$, where $K$ is the number of output classes (in our case, three: Normal, Abnormal, and Pneumonia), and $d'$ is the dimensionality of each class capsule. We start by computing a prediction vector $\hat{U}_{ij}$ for each pair of input capsule $i$ and output class capsule $j$, using a learned transformation matrix $W_{ij}$.

$$\hat{u}_{ij} = v_i \cdot W_{ij}$$
(6)

Next, we determine how strongly each input capsule $i$ contributes to each output class capsule $j$, using a coupling coefficient $c_{ij}$, calculated over the logits $b_{ij}$.

$$c_{ij} = \frac{e^{b_{ij}}}{\sum_{k=1}^{K} e^{b_{ik}}}$$
(7)

In equation 7, $b_{ij}$ represents the log prior probability (agreement score) between capsule $i$ and class $j$, and $e$ denotes the exponential function. The denominator sums over all possible classes $k$, where $k \in \{1, 2, \ldots, K\}$.

Then, using the coupling coefficients, we compute the weighted sum $s_j \in \mathbb{R}^{d'}$ of all prediction vectors directed toward class $j$, which is given by:

$$s_j = \sum_{i=1}^{N} c_{ij} \cdot \hat{u}_{ij}$$
(8)

Now, we apply the squash function once more to obtain the final output vector $v_j$ for each class given as:

$$v_j = \frac{\|s_j\|^2}{1 + \|s_j\|^2} \cdot \frac{s_j}{\|s_j\| + \in}$$
(9)

By combining the outputs for all classes, we construct the final capsule matrix $V' = [v_1, v_2, v_3] \in \mathbb{R}^{K \times d'}$, where each $v_j$ captures spatial, semantic, and agreement-based features for a specific class.

**3.3.4 Dense and dropout layers.** After obtaining the final capsule matrix $V' = [v_1, v_2, v_3] \in \mathbb{R}^{K \times d'}$, where each vector $v_j \in \mathbb{R}^{d'}$ encodes the spatially aggregated features for class $j$, we convert this capsule representation into a single high-dimensional vector. By reshaping the matrix, we obtain a feature vector $z \in \mathbb{R}^{K \cdot d'}$, which acts as input to the fully connected layers.

We pass the vector $z$ through a dense layer with a Rectified Linear Unit (ReLU) activation function, producing an intermediate representation $h \in \mathbb{R}^D$, which is given as:

$$h = ReLU(W_1 z + b_1) \tag{10}$$

Here, $W_1 \in R^{D \times (K \cdot d')}$ and $b_1 \in R^D$ are learnable parameters, and $D$ is the dimensionality of the dense layer output. To reduce overfitting, we apply dropout to the activations $h$, resulting in the masked vector $\tilde{h} \in R^D$. Finally, we compute the output vector $p \in R^K$, that consists of unnormalized classification scores for each class as:

$$p = W_2 \tilde{h} + b_2 \tag{11}$$

Here, $W_2 \in R^{K \times D}$ and $b_2 \in R^K$ are the parameters of the final classification layer. Thus, in contrast to conventional capsule networks, which use the vector norm $\|v_j\|$ as a direct measure of class probability, our method utilizes capsule outputs as representations of enhanced features. To get class-specific confidence scores, the model uses deeper transformations by projecting them across fully connected layers. The spatial and hierarchical information conveyed by the capsules is thus retained.

### 3.3.5 Fuzzy inference system for severity classification.

A collection of language expressions known as fuzzy rules explains how to use a fuzzy inference system to classify inputs or regulate outputs. The fuzzy inference system translates classification confidence scores and pneumonia symptoms into severity levels.

**Mamdani fuzzy inference.** This work utilizes a Mamdani-type fuzzy Inference System, which maps 11 pneumonia-associated symptoms/diagnoses in a clinically predefined range, and also the DL based confidence scores into Fuzzy sets using Fuzzy Membership Functions (type-1 fuzzy). The fuzzy rules used in our inference engine, as shown in Table 4, were referred from standard clinical practice guidelines for pneumonia diagnosis, including severity indices such as the CURB-65 (Confusion, Urea, Respiratory rate, Blood pressure, Age ≥ 65) score [47], PSI (Pneumonia Severity Index) [48,49], and WHO (World Health Organization) clinical case definitions [50]. These guidelines were carefully studied to extract key symptoms and combinations relevant to pneumonia severity.

Here, most clinical factors have values between 0–1, which represents the severity scale, even for the ones that are clinically defined as "*yes it's present*" or "*no it's absent*". But conditions such as fever value, oxygen level, and fever duration require actual parameters. So, we applied normalization to ensure that all inputs contribute equally to the fuzzy system without being biased by their original units or magnitudes.

As shown in equation 12, $\Delta$ refers to the actual input, $\Delta_{min}$ refers to the minimum boundary, $\Delta_{max}$ is the maximum boundary, and $\theta$ refers to the normalized value.

$$\theta = \frac{\Delta - \Delta_{min}}{\Delta_{max} - \Delta_{min}} \tag{12}$$

The system categorizes the fuzzy membership values into linguistic labels such as "Poor," "Average," and "Good". These membership categories are crucial for transforming crisp input values into fuzzy representations, forming the basis for rule-based reasoning. This implementation uses piecewise linear membership functions to determine fuzzy grades based on symptom severity.

**Table 4. Input membership ranges assigned to associated symptoms and conditions used in the fuzzy inference system for pneumonia severity classification.**

| Associated symptoms/conditions | Input membership range | Clinical Guideline Source |
|---|---|---|
| breathlessness, sputum production, hemoptysis, fatigue, appetite loss, chest pain, cough severity, confusion | (0-1) | WHO, CURB-65, PSI |
| fever value, oxygen level, fever duration | (35.0-42.0), (80.0-100.0), (0.0-30.0) | WHO, PSI |

Each input symptom x is transformed into its corresponding fuzzy membership grade $\mu_x(x)$ using the piecewise linear functions, where x represents the crisp symptom value, min and max denote the range of possible symptom values. Fig 6 illustrates how pneumonia symptoms are mapped into fuzzy categories—poor, average, and good—using triangular/trapezoidal membership functions. Each symptom is assigned a membership value based on its normalized severity, allowing for a smooth transition between categories.

The "Poor" function decreases as severity increases, "Good" increases with severity, while "Average" peaks in between, covering intermediate values. These functions are critical for the fuzzy logic-based pneumonia diagnosis, as the membership category is determined, as shown in Table 5.

**Rule activation and aggregation.** Expert-defined fuzzy rules are activated based on input variables, such as IF-THEN statements, which evaluates multiple patient symptoms to determine the pneumonia severity score. Logical operators [51] 'AND' and 'OR' are used to establish the final severity level, as shown in Table 6, ensuring robust classification.

While fixed operator-level scores are rarely assigned explicitly in published fuzzy systems, similar effects are achieved by assigning weighted contributions to activated fuzzy rules. Indeed, some studies demonstrate additive rule aggregation in pneumonia and stroke diagnosis [52,53].

Fig 7 highlights the overall fuzzy rule-based pneumonia severity classification using symptom combinations and membership contributions. The associated symptoms and clinical conditions were developed in co-ordination with an expert Cardiologist. Details of the clinician's notes are provided in the supporting information section of this paper.

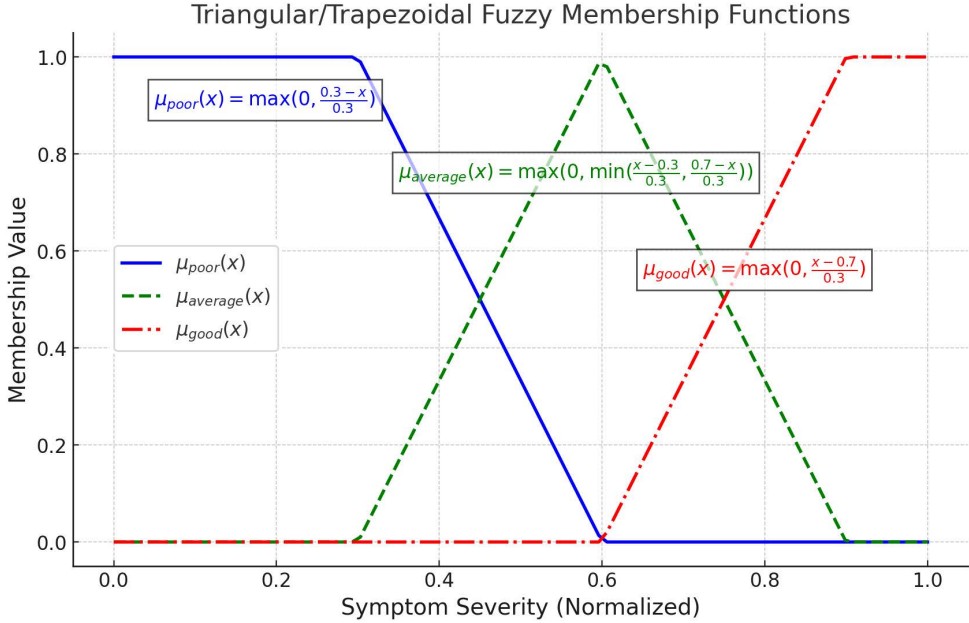

**Fig 6. Fuzzy membership functions depending upon symptom severity.**

**Table 5. Symptom classification by fuzzy membership ranges (poor, average, and good).**

| Membership category | Range (Normalized values) |
| --- | --- |
| Poor | x < 0.3 |
| Average | 0.3 ≤ x ≤ 0.7 |
| Good | x > 0.7 |

**Table 6. Fuzzy rule activation logic based on logical operators.**

| Logic Type | Activation Condition | Interpretation | Score Contribution | Example |
|---|---|---|---|---|
| AND | All symptom conditions in the rule must be satisfied | Indicates a strict rule where every symptom must meet its threshold simultaneously. | +10 | IF breathlessness is good AND fever is average AND sputum is poor →THEN severity is Severe Pneumonia |
| OR | At least one symptom condition must be satisfied | Allows flexibility; the rule activates even if only one symptom meets its condition. | +5 | IF Appetite Loss is average OR Fever value is good →THEN severity is Moderate Pneumonia |

All conditions must be satisfied simultaneously in a rule for it to be triggered. Since multiple rules exist, the final aggregated membership for pneumonia severity $\mu_{Severity}$ is taken as the maximum of all activated rules, which is given by:

$$\mu_{Severity} = \sum_{k=1}^{m} w_k \times \mu_k$$

(13)

Here, $m$ represents the total number of fuzzy rules. $w_k$ is the weight assigned to rule $k$ based on logic type AND operator and OR operator, which yield +10 points and +5 points respectively. $\mu_k$ is the membership activation of the $k^{th}$ rule. To ensure the stability and accuracy of severity classification. CNN confidence scores are integrated with the fuzzy rule-based severity estimator.

**Initial defuzzification.** To make actionable clinical decisions, the fuzzy output $\mu_{severity}$ must be converted into a crisp value through defuzzification. In this work, we adopt the weighted average defuzzification method, where the final pneumonia severity score $S_{fuzzy}$ is computed as:

$$S_{fuzzy} = \frac{\mu_{severity}}{\sum_{k=1}^{m} w_k}$$

(14)

**Final defuzzification with DenseNet confidence score**. By applying the DenseNet confidence as a proportionate weighting factor, the fuzzy severity score is constantly adjusted. After preprocessing the chest X-ray picture with CLAHE to improve local contrast, the image is normalized and resized. For hierarchical feature extraction, the preprocessed image is run through a deep convolutional backbone (DenseNet201). The spatial linkages and feature orientation are then maintained by feeding these features into a Capsule Network made up of Primary layers and dynamic routing layers. Following flattening and passing through dense layers, the final capsule output yields three confidence values that represent the Normal, Abnormal, and Pneumonia classes.

Let the DL's output probability vector be $\boldsymbol{p} = [p_1, p_2, p_k]$, where $p_i \in [0,1]$ denotes the probability for class $i$, and $k$ represents the number of classes. Since we have three classes (Abnormal, Normal and Pneumonia), the probability distribution over $k$ classes is given by:

$$\sum_{i=1}^{k} p_i = 1$$

(15)

To improve interpretability and reduce overconfidence in raw SoftMax results, we combine class separation, entropy-based uncertainty, and clinical weighting relevant to each class to calculate a calibrated confidence score. Let $p_{top1}$ and $p_{top2}$ be the highest and second-highest probabilities in the prediction vector $\boldsymbol{p}$, respectively. We define margin $M$ between the two most likely classes as given by:

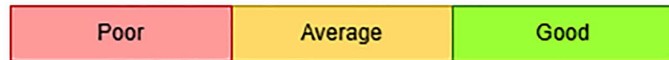

**Fig 7. Fuzzy logic-based pneumonia severity classification using membership contribution.**

$$M = p_{top1} - p_{top2} \tag{16}$$

Next, we calculate the entropy **H** (to capture uncertainty) of the prediction vector of the prediction vector **p** and a small constant $\in$ to avoid numerical instability. This is given by:

$$H(\boldsymbol{p}) = -\sum_{i=1}^{k} p_i.log(p_i + \in) \tag{17}$$

Now, we normalize the entropy ($H_{norm}$) into the range [0,1] using the maximum entropy $log(k)$, which is given by:

$$H_{norm} = \frac{H(\boldsymbol{p})}{log(k)} \tag{18}$$

Next, we calculate the calibrated confidence score $C$ as a product of the top probability, the margin, and the inverse of normalized entropy, given by:

$$C = p_{top1} \cdot M \cdot e^{-H_{norm}} \tag{19}$$

Again, we apply a class-specific weighting factor apply a class-specific weighting factor $\lambda$ that represents the clinical relevance of each class. Let relevance scores be $r$ = [0.25, 0.50, 0.75], and $a$ be the predicted class index. Then, Clinically Weighted Softmax Scaling (CWSS) is calculated as:

$$\lambda = \frac{e^{r_a}}{\sum_{i=1}^{3} e^{r_i}} \tag{20}$$

Furthermore, the contribution factor *Confidence* is calculated as:

$$Confidence = 100 * \lambda * C \tag{21}$$

Finally, the CNN confidence contribution is computed as shown by:

$$S_{final} = S_{fuzzy} + Confidence \tag{22}$$

This method ensures that higher fuzzy severity scores contribute more to DenseNet confidence, thereby reinforcing severe classifications. When total severity scores are low, the confidence score has minimal influence, preventing overestimation. The final severity classification is determined by the highest adjusted severity score, balancing contributions from fuzzy rules and DenseNet confidence, which is shown in Table 7.

**3.3.6 Final severity output.** The system further classifies pneumonia into four severity levels: Negligible Pneumonia, Mild Pneumonia, Moderate Pneumonia, and Severe Pneumonia.

Negligible Pneumonia cases have few activated rules, leading to a low severity score. Mild Pneumonia falls within a score range of 10–25, indicating symptoms that are present but not severe. Moderate Pneumonia scores 25–40, reflecting multiple symptoms without extreme severity, while Severe Pneumonia exceeds 40, with strong fuzzy rules activated. CNN confidence enhances classification consistency, reinforcing severity determination. Key symptoms like breathlessness, low oxygen levels, and high fever play a critical role in diagnosis. The final severity classification is based on the highest adjusted score after applying CNN confidence, ensuring a robust and adaptable clinical assessment.

**Table 7. Final pneumonia severity classification based on rule-based scoring.**

| Severity level (Pneumonia) | Rule-based score range |
|---|---|
| Negligible | Total severity score < 10 |
| Mild | 10 ≤ Score < 25 |
| Moderate | 25 ≤ Score < 40 |
| Severe | Score ≥ 40 |

## 3.4 Dynamic membership adjustment algorithm

This work has integrated Dynamic Fuzzy membership adjustment to cover the problem of biased Diagnosis from the previous static models. The symptom member values, which are recorded over time, undergo the following algorithm to generate a new adjusted membership range.

**Step 1:** Check the Data Frame for Trending Symptoms, then perform Data cleaning by filtering out values that deviate by more than three standard deviations from the mean using Z-Score [54]. This is given by:

$$Z-score = \frac{Value - Mean}{Standard\ Deviation} \tag{23}$$

**Step 2:** Calculate recent variance $\sigma^2$ and Mean $\mu$ for the last 10 data points ($n = 10$) as:

If data points < 10, use the variance and mean of all available data

$$\sigma^2 = \frac{1}{n} \sum_{i=1}^{n} (x_i - \mu)^2 \tag{24}$$

**Step 3:** Calculate the smoothing factor $S$ with initial base smoothing $B$ defined as 0.3, given by:

$$S = B + \frac{\sigma^2}{100} \tag{25}$$

**Step 4:** Apply the exponential smoothing [55]

  **4.1** Initialize the first value of the weighted symptom data

  **4.2** For each symptom data point, $x_i$ $S_{x_i}$ as:

$$S_{x_i} = (S \times x_i) + (1 - S) \times x_{i-1} \tag{26}$$

**Step 5:** Set percentile threshold values for the recent mean value $\mu$

  **5.1** If $\mu < 0.5$; set lower percentile = 10% and upper percentile = 90%

  **5.2** If $\mu > 0.5$; set lower percentile = 15% and upper percentile = 85%

**Step 6:** Calculate adjusted minimum and maximum range as per the percentile thresholds values, then generate adjusted range $A_r$ of values from the adjusted minimum to the adjusted maximum with an increment of 0.1, as given by:

$$A_r = [A_{min}, A_{min} + 0.1, \ldots, A_{max}] \tag{27}$$

**Step 7:** Limit the adjusted range $A_r$ within the specified minimum and maximum bounds and return the clipped range

## 4. Results

### 4.1 Simulation environment

The system, powered by an Intel Core i7 10700k CPU, 16GB RAM, and an NVIDIA RTX 3070 GPU, was designed for efficient deep learning tasks. It was developed using Windows 10 with Visual Studio Code, Sublime Text Editor, Jupyter Notebook, and Python as the primary programming language.

### 4.2 Libraries used

TensorFlow was used to define and load deep learning models, such as Capsule Network and CNN models. This library is the core of Data augmentation and image preprocessing methods used for model training, and was trained efficiently to classify the CXR images [56,57]. It was also used to build custom layers that implemented the Capsule Network function. NumPy [58] was used for array based operations like flattening model features, Image Preprocessing operation (CLAHE), image normalization and calculating fuzzy input factors for fuzzy logic classification. The fuzzy membership values assigned to symptoms were calculated dynamically using NumPy operations, ensuring smooth integration between deep learning outputs and the rule-based fuzzy inference system. Matplotlib was used for plotting training accuracy and Validation loss plots [59], and specifically for plotting Gradient-weighted Class Activation Mapping (Grad-CAM) heatmaps.

### 4.3 Results of cross-validation methods for pneumonia classification

To identify the most suitable model for pneumonia diagnosis, we evaluated three DenseNet architectures (DenseNet121, DenseNet169, and DenseNet201) using four validation methods: Stratified k-Fold, Monte Carlo, k-Means, and Bootstrap. As mentioned in the methodology section, the test dataset remained constant across all the folds of these cross-validation techniques. In each of the 5 folds, we specifically focused on validation loss. Our model stops training if the loss fails to improve for 5 times in a row (known as early stopping), despite reducing the learning rate by a small factor.

As per Table 8, DenseNet201 was chosen for consistent performance across multiple cross-validation methods. Unlike DenseNet121 and DenseNet169, DenseNet201 maintained low validation loss and good accuracy with minimal variance across Stratified K-Fold, Monte Carlo, K-Fold, and Bootstrap validations. Therefore, we selected the Stratified k-fold-based method due to good generalization on both the test and validation dataset with 99.01% and 97.93% accuracy respectively.

As per the training graphs shown in Fig 8, DenseNet201 in the 5th fold (stratified k-fold) showed efficient convergence, reducing overfitting and narrowing the accuracy gap between training and validation. Its balanced performance indicated strong generalization, while metrics like accuracy difference provided deeper insights into model behavior. Also, Fig 9 represents the confusion matrix for each of the five folds. Furthermore, we calculated the *precision*, *recall*, *F1-score* and *specificity* for the test dataset based on True Positive (*TP*), False Negative (*FN*), False Positive (*FP*) and True Negative (*TN*), given by equation 28–31. We have illustrated the individual metrics on Table 9.

$$Precision = \frac{TP}{TP + FP} \tag{28}$$

$$Recall = \frac{TP}{TP + FN} \tag{29}$$

**Table 8. Lowest validation loss achieved from one of the 5 folds/iterations with corresponding metrics.**

| Model | Cross-validation method | Fold/ Iteration number | Lowest Validation loss (%) | Validation accuracy (%) | Training accuracy (%) | Training loss (%) | Test Accuracy (%) |
|---|---|---|---|---|---|---|---|
| Dense121 | Stratified k-fold | 3 | 5.91% | 98.33% | 98.52% | 5.19% | 98.34% |
| | Monte Carlo | 2 | 7.11% | 97.66% | 97.72% | 7.38% | 98.87% |
| | k-fold | 4 | 6.61% | 98.06% | 97.72% | 7.78% | 98.25% |
| | Bootstrap | 5 | 7.07% | 97.86% | 97.55% | 7.79% | 98.92% |
| Dense169 | Stratified k-fold | 3 | 5.64% | 98.33% | 97.32% | 8.58% | 98.74% |
| | Monte Carlo | 5 | 7.26% | 97.76% | 97.87% | 7.27% | 99.01% |
| | k-fold | 3 | 6.93% | 97.83% | 98.19% | 6.00% | 99.01% |
| | Bootstrap | 3 | 6.49% | 82.03% | 94.47% | 16.50% | 93.67% |
| Dense201 | Stratified k-fold | 5 | 6.21% | 97.93% | 98.04% | 6.46% | 99.01% |
| | Monte Carlo | 3 | 7.63% | 97.49% | 97.80% | 7.07% | 98.96% |
| | k-fold | 3 | 5.64% | 98.30% | 98.20% | 5.97% | 99.14% |
| | Bootstrap | 3 | 7.80% | 97.59% | 97.89% | 6.77% | 98.65% |

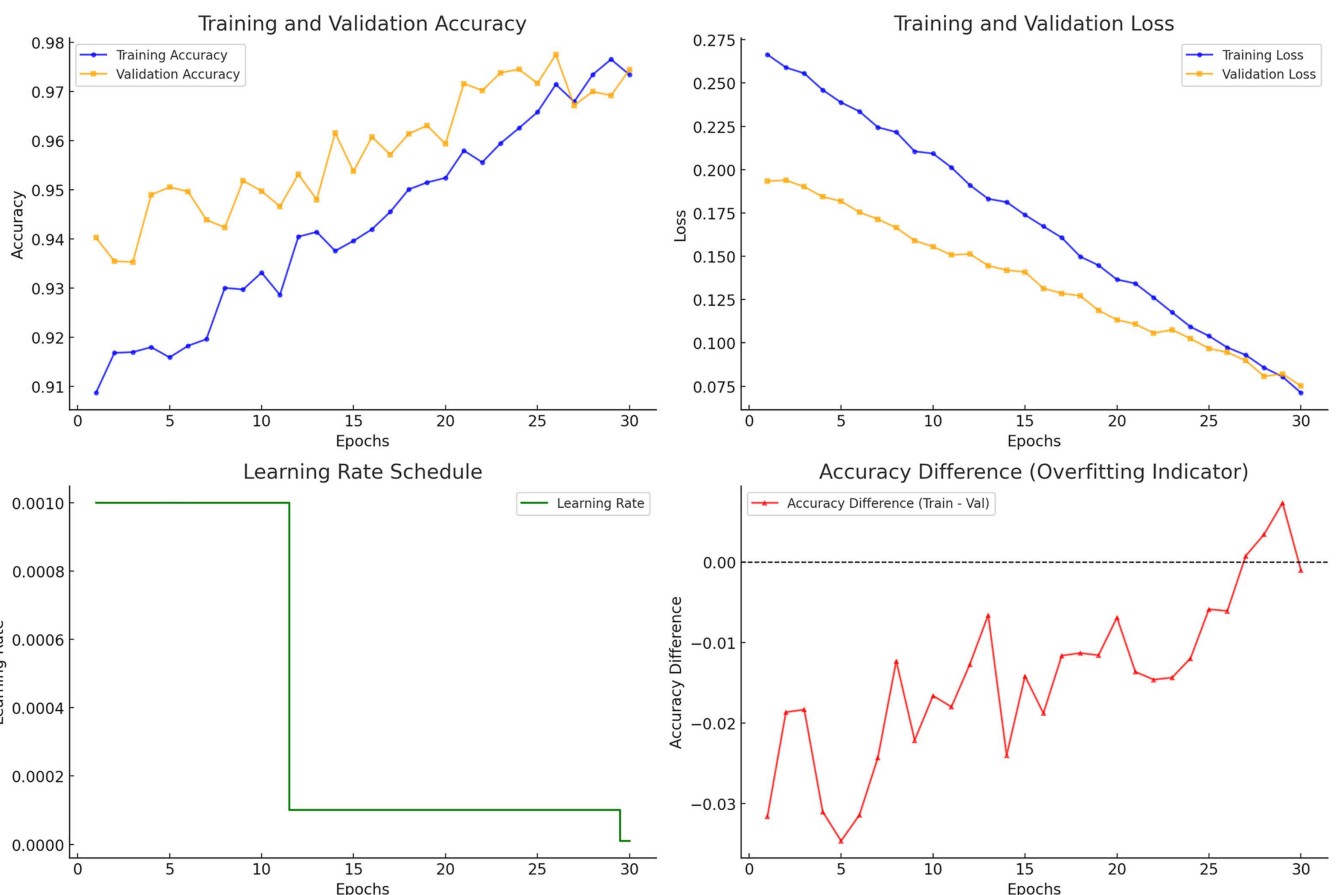

**Fig 8. Training performance metrics for DenseNet201 – Fold 5: accuracy and loss trends over epochs, learning rate adjustments, and accuracy difference indicating generalization behavior.**

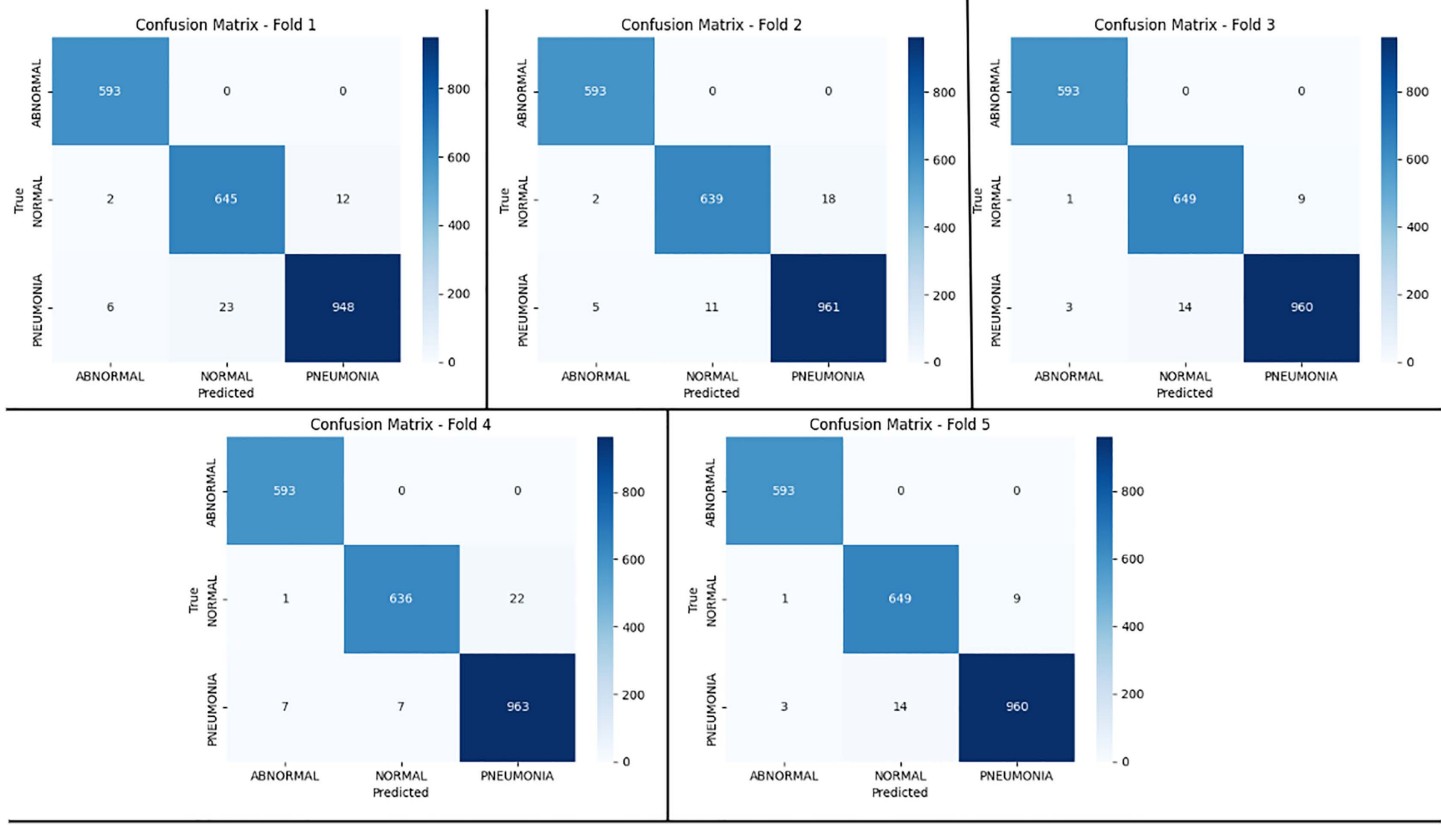

**Fig 9. Confusion matrix from each of the five folds of the stratified k-fold test dataset based on Dense201+CapsNet model.**

**Table 9. Per-class evaluation metrics across Stratified k-folds for DenseNet201 model on test dataset.**

| Fold | Class | Precision | Recall | F1-Score | Specificity |
|---|---|---|---|---|---|
| Fold 1 | ABNORMAL | 0.9867 | 1 | 0.9933 | 0.9951 |
| | NORMAL | 0.9656 | 0.9788 | 0.9721 | 0.9854 |
| | PNEUMONIA | 0.9875 | 0.9703 | 0.9788 | 0.9904 |
| Fold 2 | ABNORMAL | 0.9883 | 1 | 0.9941 | 0.9957 |
| | NORMAL | 0.9831 | 0.9697 | 0.9763 | 0.993 |
| | PNEUMONIA | 0.9816 | 0.9836 | 0.9826 | 0.9856 |
| Fold 3 | ABNORMAL | 0.9933 | 1 | 0.9966 | 0.9976 |
| | NORMAL | 0.9789 | 0.9848 | 0.9818 | 0.9911 |
| | PNEUMONIA | 0.9907 | 0.9826 | 0.9866 | 0.9928 |
| Fold 4 | ABNORMAL | 0.9867 | 1 | 0.9933 | 0.9951 |
| | NORMAL | 0.9891 | 0.9651 | 0.977 | 0.9955 |
| | PNEUMONIA | 0.9777 | 0.9857 | 0.9817 | 0.9824 |
| Fold 5 | ABNORMAL | 0.9933 | 1 | 0.9966 | 0.9976 |
| | NORMAL | 0.9789 | 0.9848 | 0.9818 | 0.9911 |
| | PNEUMONIA | 0.9907 | 0.9826 | 0.9866 | 0.9928 |

$$F1 - score = 2 \times \frac{Precision \times Recall}{Precision + Recall} \tag{30}$$

$$Specificity = \frac{TN}{TN + FP} \tag{31}$$

It is mentioned earlier that we preferred the performance metrics of the 5th fold (99.01% test accuracy). The model achieved high overall performance, with a micro-averaged precision, recall, and F1-score of 0.9879. Macro-averaged scores were 0.9876 for precision, 0.9891 for recall, and 0.9883 for F1-score. Similarly, weighted precision was 98.78%, a weighted recall was 98.74%, and a weighted F1-score was 0.9876.

**4.3.1 Comparison with PELM.** To put our DL results in perspective, we also looked at a recent pneumonia model called the Pneumonia Ensemble Learning Model (PELM) by Yanar et al. [60]. This model was designed for binary pneumonia detection using chest X-rays. Their system combines several well-known deep learning architectures, including InceptionV3, VGG16, ResNet50, and Vision Transformer (ViT). It was trained on 50,000 images. On their set of 5,000 test samples, they reported strong results: 96% accuracy, 99% precision, 91% recall, and an F1-score of 0.95.

Our system, on the other hand, was evaluated using a more rigorous approach, Stratified k-Fold cross-validation, which ensured our model performed consistently across different data splits. To repeat, in the 5th fold of Stratified k-Fold cross-validation using DenseNet201, our model achieved an overall test accuracy of 99.01%, with a weighted precision of 98.78%, a weighted recall of 98.74%, and a weighted F1-score of 0.9876. These results demonstrate strong and consistent classification performance across all three classes. It is evident that CapsNet also enhanced accuracy, which has been discussed later in this work.

But the key difference goes beyond just the numbers. While PELM focuses on binary classification—pneumonia or not—our system offers an additional "ABNORMAL" category, which is essential for the identification of early Pneumonia. Furthermore, this research offers Grad-CAM visualizations, which have been noted as a future scope of PELM, making our model clinically interpretable as well.

## 4.4 Significance of capsule layers

To validate the robustness of the Capsule Network, we trained a DenseNet201 model with a standard Fully Connected (FC) layer using the same Stratified k-fold split.

Alternatively, the investigation involved five bootstrap sampling iterations, randomly selecting samples equal to the test set. The statistical significance of performance differences between the two models was determined using mean, standard deviation, 95% confidence intervals, and a paired t-test. The results achieved in Table 10 confirm that Capsule Network continuously performed better than the mainstream FC model in every metric, in terms of accuracy (0.9876 vs. 0.9821),

**Table 10. Comparative evaluation of CapsNet and FC models using bootstrap sampling across five test folds, showing performance metrics with 95% confidence intervals and p-values.**

| DL Pipeline | Accuracy (±SD)/ CI | Precision (±SD)/ CI | Recall (±SD)/ CI | F1 Score (±SD)/ CI | Specificity (±SD)/ CI | ROC AUC (±SD) |
|---|---|---|---|---|---|---|
| CapsNet | 0.9876±0.0020, CI:0.9859–0.9894 | 0.9873±0.0015, CI:0.9859–0.9886 | 0.9889±0.0022, CI:0.9870–0.9909 | 0.9881±0.0018, CI:0.9865–0.9897 | 0.9937±0.0012, CI: 0.9926–0.9947 | 0.9995±0.0001, CI:0.9993–0.9996 |
| FC | 0.9821±0.0011, CI:0.9811–0.9832 | 0.9810±0.0007, CI:0.9804–0.9816 | 0.9845±0.0013, CI:0.9834–0.9856 | 0.9827±0.0009, CI:0.9818–0.9835 | 0.9912±0.0008, CI:0.9905–0.9918 | 0.9991±0.0001, CI:0.9990–0.9992 |
| **paired t-test (p-value)** | 0.0066 | 0.0029 | 0.0203 | 0.0071 | 0.0137 | 0.0012 |

precision (0.9873 vs. 0.9810), recall (0.9889 vs. 0.9845), F1 score (0.9881 vs. 0.9827), specificity (0.9937 vs. 0.9912), and ROC AUC (0.9995 vs. 0.9991). All improvements were statistically significant with p-values less than 0.05, indicating the superior performance of the Capsule-enhanced architecture.

### 4.5 Grad-CAM based visualization

 represents the Grad-CAM based visualization used for pneumonia classification, providing a clearer understanding of how AI systems make clinical decisions. The model shows predominant activity in the lower and central lung fields, suggesting it has learned to focus on areas with pulmonary consolidation. The heatmap used for the mask's outlines aligns with anatomical features, allowing visual validation of the model's gaze regions. The binary mask produced using Otsu's thresholding method captures high activity values and transforms them into discrete areas that mark important activities. This allows for the confirmation of localized pneumonia suspension, and can be beneficial for achieving clinical significance.

### 4.6 Stabilization of the dynamic membership adjustment algorithm

The fuzzy membership stabilization system enhances pneumonia severity classification by dynamically adjusting symptom boundaries over time. It refines limits for the associated pneumonia symptoms/conditions through evaluations and smoothing methods. We tested the effectiveness of the algorithm by performing stabilization tests at iterations 10, 20, 30, 50, and 100. With each iteration, new symptom values update membership bounds (Bounds used in the next iteration reflect the updated values from the previous ones).

A damping factor reduces large variations, enhancing adaptability. To filter outliers, a Z-score method is used, preventing extreme values from impacting adjustments. Additionally, an exponential smoothing algorithm is applied, which adjusts

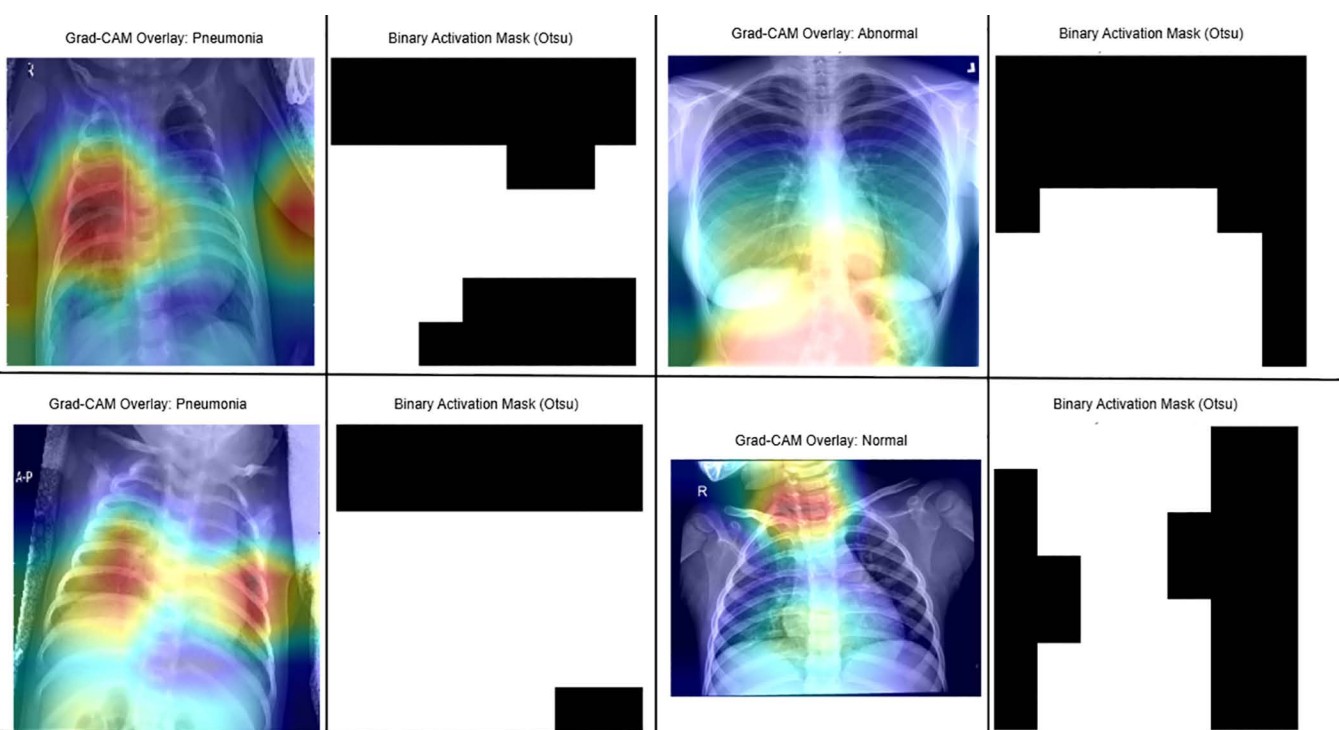

**Fig 10. Grad-CAM heatmap highlighting affected regions as per the predicted class (left), and its corresponding binary activation mask using Otsu thresholding (right) to extract high-importance areas for explainability and localization.**

the weight of recent symptom values based on variance and thresholds for gradual updates. To measure effectiveness, fluctuation percentages are calculated at different steps, and the reduction over time is assessed. Stabilization accuracy is determined by comparing fluctuations from 10 to 100 iterations. To quantify stabilization, the system computes initial and final averages of membership fluctuation, as shown in equation 32 and equation 33.

$$Initial\ Average = \frac{Mean\ Initial\ Mimimum + Mean\ Initial\ Maximum}{2} \tag{32}$$

$$Final\ Average = \frac{Mean\ Final\ Mimimum + Mean\ Final\ Maximum}{2} \tag{33}$$

Where *Mean Initial Minimum* and *Mean Initial Maximum* are the average minimum and maximum bound fluctuations at the 10th iteration, respectively, while *Mean Final Minimum* and *Mean Final Maximum* represent the average minimum and maximum bound fluctuations at the 100th iteration. If any symptom fluctuates more than 1% at 100 iterations, additional stabilization steps are triggered to further refine the fuzzy membership boundaries. The stabilization factor (*SF*), which ensures controlled adjustments, is calculated as:

$$SF = \max\left(0.4,\ 1 - \frac{Final\ Average}{Initial\ Average}\right) \tag{34}$$

The fluctuation reduction (*F*), which measures the percentage decrease in fluctuation over iterations, is given by:

$$F = \frac{Initial\ Average - Final\ Average}{Initial\ Average} \times 100 \tag{35}$$

As shown in Table 11, the system effectively reduces fluctuations in fuzzy membership adjustments, ensuring reliable and consistent pneumonia symptom tracking for healthcare practitioners. An instance of the stabilization test is shown in Fig 11 earlier, highlights that the adaptive fuzzy adjustment method effectively maintains consistent and reliable symptom representation through stable membership bounds.

**Table 11. Evolution of Minimum and Maximum Membership Bounds for Each Symptom Across Iteration Steps.** The table demonstrates the modified lower and upper boundaries of symptom membership for pneumonia across 11 symptoms during steps 20, 50 and 100. The figures represent final boundaries obtained through dynamic adjustment in each step, thus showing how symptom membership ranges change and reach stability over time because of input patterns. The boundaries were chosen from the final step of every process, which demonstrates how the fuzzy system approaches stability.

| Symptom | Min/Max @10 | Min/Max @20 | Min/Max @30 | Min/Max @50 | Min/Max @100 |
|---|---|---|---|---|---|
| Breathlessness | 0.48/ 0.92 | 0.11/ 0.81 | 0.04/ 0.80 | 0.32/ 0.79 | 0.07/0.80 |
| Sputum production | 0.04/ 0.70 | 0.16/0.78 | 0.04/ 0.83 | 0.15/ 0.72 | 0.1/ 0.83 |
| Fever Duration | 3.08/ 22.80 | 9.66/28.63 | 17.16/27.1 | 2.66/27.656 | 8.64/24.13 |
| Fever Value | 35.72/39.06 | 35.78/ 40.80 | 35.85/41.27 | 37.656/ 41.49 | 36.88/ 40.98 |
| Hemoptysis | 0.40/ 0.92 | 0.21/0.79 | 0.14/ 0.78 | 0.04/ 0.81 | 0.05/ 0.58 |
| Fatigue | 0.06/ 0.76 | 0.096/ 0.77 | 0.19/ 0.92 | 0.24/ 0.83 | 0.08/ 0.81 |
| Appetite loss | 0.22/ 0.56 | 0.32/ 0.66 | 0.11/ 0.84 | 0.09/ 0.79 | 0.07/ 0.74 |
| Confusion | 0.20/ 0.59 | 0.03/ 0.72 | 0.21/ 0.96 | 0.08/ 0.77 | 0.07/ 0.74 |
| Chest pain | 0.43/ 0.85 | 0.25/ 0.76 | 0.09/ 0.73 | 0.21/ 0.81 | 0.27/ 0.94 |
| Cough severity | 0.17/ 0.58 | 0.25/ 0.89 | 0.41/ 0.91 | 0.14/ 0.83 | 0.20/ 0.64 |
| Oxygen level | 86.88/ 97.40 | 84.41/ 97.83 | 82.15/ 97.15 | 91.29/ 96.32 | 82.24/ 95.14 |

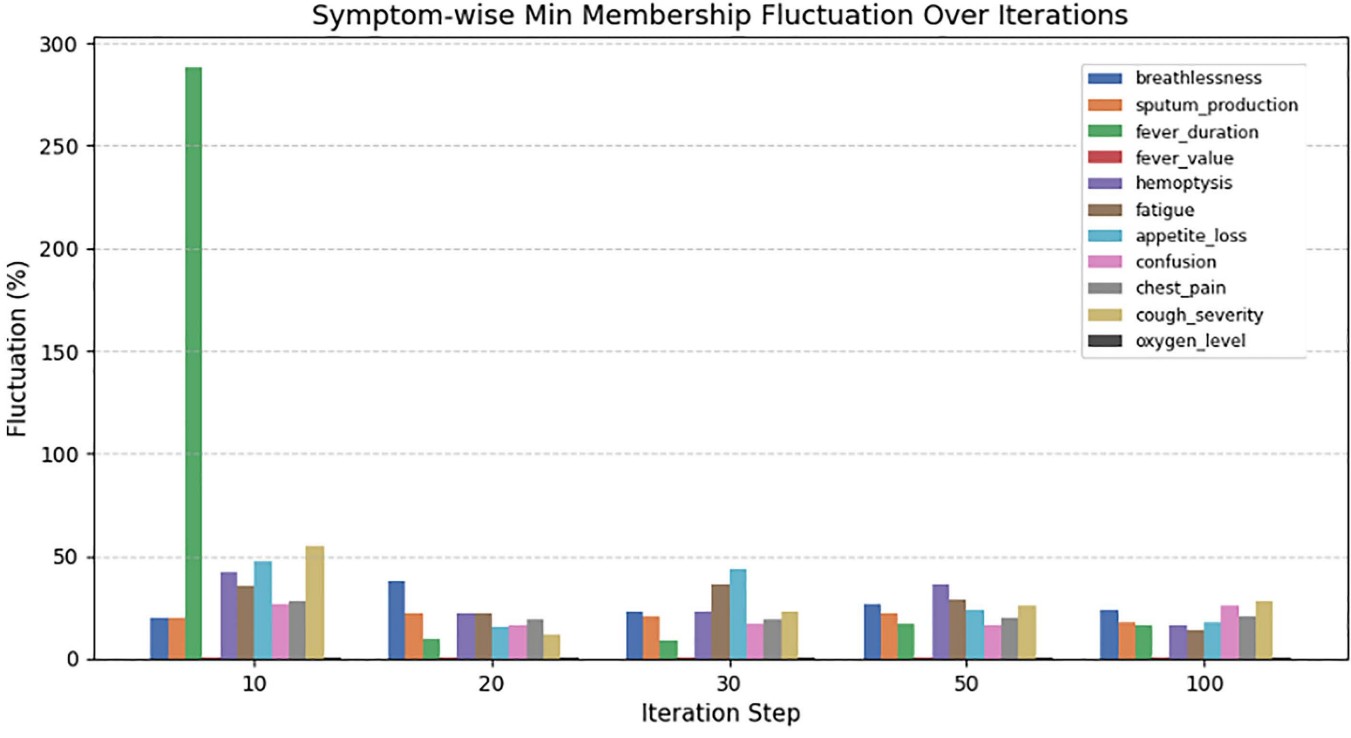

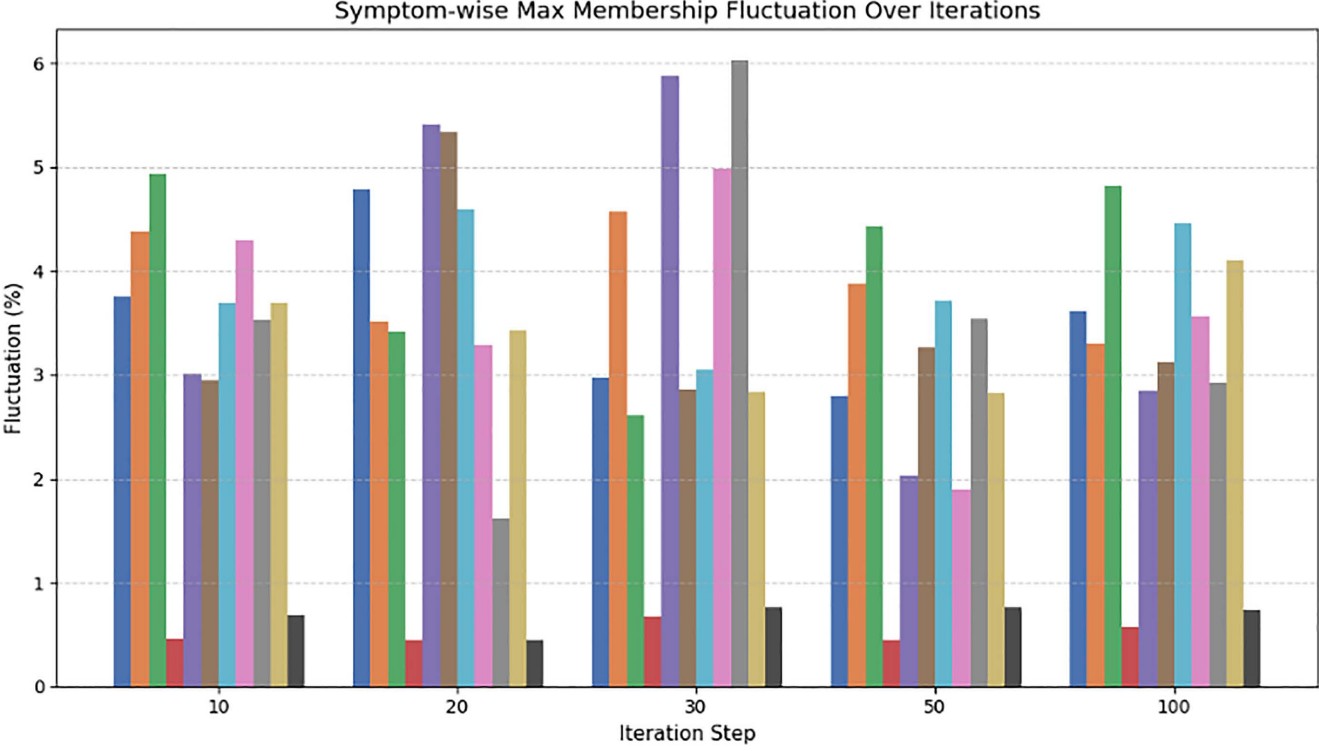

**Fig 11. Symptom-wise Minimum and Maximum Membership Fluctuation over Iterations.** The figure illustrates the dynamic behavior of fuzzy membership bounds for each symptom across multiple stabilization steps (10, 20, 30, 50, and 100 iterations). The first subplot depicts the percentage change in the minimum membership bounds graphically and we can see a very high degree of volatility at the beginning, especially for fever duration where it

surges to almost 290 percent at iteration 10. However, with further progress in iterations, the fluctuations reduce significantly for all symptoms demonstrating that there is greater consistency in the estimation of the lower bound. The second subplot shows the changes of maximum membership bounds which do not exceed 6 percent overall indicating a lower degree of variability in upper bound adjustments. Overall, the plots illustrate that the proposed adaptive fuzzy adjustment approach is capable of reliably representing the symptoms in terms of their bounds over time.

Furthermore, the standard deviation values for both minimum and maximum membership fluctuations were calculated at each iteration step (10, 20, 30, 50, and 100) using all recorded changes within that step. Let $x_1$, $x_2$, ..., $x_n$ be the list of fluctuation values at a particular step. The standard deviation σ is calculated as:

$$\sigma = \sqrt{\frac{1}{n} \sum_{i=1}^{n} (x_i - \mu)}$$

(36)

Here, $\mu$ is the mean of the fluctuation values, $n$ is the number of recorded values for that iteration step, and $x_i$ is each recorded min or max percentage change. These values measure the variability of symptom-related adjustments over time. They help evaluate the stability and consistency of the fuzzy membership updates. The results of these values are shown in Table 12.

This shows that some symptoms, like fever duration and hemoptysis, fluctuate a lot in the early stages, while others, like fever value, stay pretty stable throughout. It means that the system doesn't treat every symptom the same when it comes to how quickly or smoothly they settle down. To get more reliable results, we may need to give extra attention to the more unstable ones by letting them stabilize longer or tweaking how we update their values.

## 4.7 Influence of confidence score on pneumonia membership range

The dynamic membership adjustment algorithm allows symptoms to evolve, but it can sometimes lead to borderline classifications (a situation that occurs when the top two severity scores are nearly identical). The DL confidence serves as a moderating factor and strengthens the determination of severity.

We ran the fuzzy logic system on 300 cases to evaluate this effect, using both DL-assisted and non-DL-assisted setups. For each case, we generated symptom inputs based on a predefined severity category. Then, fuzzy rules calculated

**Table 12. Standard deviation of minimum and maximum membership fluctuation percentages for each symptom over five iteration checkpoints (10, 20, 30, 50, and 100). Values are shown as "Min StdDev/ Max StdDev," which highlights the level of variability in fuzzy bound adjustments during the stabilization process. Higher deviations in some symptoms, like fever duration and hemoptysis, show more instability in early iterations. In contrast, symptoms like fever value remain stable throughout.**

| Symptom | StdDev@10 (min/max) | StdDev@20 (min/max) | StdDev@30 (min/max) | StdDev@50 (min/max) | StdDev@100 (min/max) |
|---|---|---|---|---|---|
| Breathlessness | 44.65/ 13.43 | 52.14/ 4.25 | 24.63/ 8.48 | 40.11/ 4.42 | 55.92/ 6.79 |
| Sputum production | 14.07/ 7.68 | 37.03/ 8.50 | 60.12/ 4.33 | 73.60/ 5.90 | 76.31/ 7.40 |
| Fever duration | 1124.50/ 9.26 | 48.47/ 6.14 | 24.23/ 9.18 | 30.32/ 8.63 | 22.07/ 6.96 |
| Fever value | 0.95/ 0.95 | 0.74/ 0.71 | 0.75/ 0.95 | 0.92/ 0.69 | 0.90/ 0.91 |
| Hemoptysis | 562.44/ 5.93 | 28.46/ 3.51 | 28.14/ 7.81 | 55.15/ 7.64 | 66.95/ 6.25 |
| Fatigue | 153.97/ 6.28 | 35.69/ 6.45 | 31.63/ 6.19 | 45.42/ 5.27 | 48.16/ 8.14 |
| Appetite loss | 405.05/ 6.75 | 40.69/ 4.49 | 30.08/ 3.93 | 31.09/ 12.51 | 44.93/ 9.15 |
| Confusion | 178.46/ 5.18 | 45.82/ 10.24 | 62.67/ 8.07 | 34.45/ 4.49 | 75.64/ 6.52 |
| Chest pain | 189.10/ 4.79 | 34.97/ 3.45 | 36.27/ 8.37 | 40.52/ 8.35 | 51.83/ 5.92 |
| Cough severity | 169.29/ 6.50 | 28.07/ 7.59 | 27.88/ 8.31 | 40.98/ 6.97 | 33.37/ 7.46 |

the severity scores with and without the DL confidence values. We flagged a case as borderline if the difference between the top two severity scores, also known as the margin, was less than or equal to 2.0 in either setup.

For each borderline case, we calculated the difference in severity scores between DL and non-DL settings across all four severity classes: severe, moderate, mild, and negligible. These score differences helped us understand how much the CNN confidence influenced the fuzzy system. The results were visualized class by class using individual line plots, as shown in Fig 12. This clearly showed that DL based confidence had a strong impact in some cases, particularly in moderate cases, while in others, such as severe cases, the scores remained mostly unchanged.

This method helps explain where DL based guidance is most important and supports clear, data-driven reasoning behind severity decisions in pneumonia diagnosis.

## 5. Conclusion and future work

This study addresses a significant gap in pneumonia diagnosis — the inability of most models to identify ambiguity in Pneumonia diagnosis via the use of DL based confidence score, generated from CXR, or anything similar. We built a hybrid AI model that not only diagnoses chest X-ray images with DenseNet, but also interprets symptom severity using a fuzzy logic system that dynamically adapts according to the reported severity. By combining the DL confidence score with fuzzy reasoning, our system improves handling the vagueness of borderline and uncertain cases, which static models typically are not capable of doing. The DenseNet with CapsNet for Stratified k-fold validation scored 99.01% in the test set, and recorded a validation loss of 6.21%, significantly better than other models and validation methods. Furthermore, the final model achieved a weighted precision of 0.9878, weighted recall of 0.9874, and a weighted F1-score of 0.9876, confirming its robust performance across all classes.

Upon further test, the CapsNet-based pipeline outperformed the standard FC-based pipeline in terms of accuracy (0.9876 vs. 0.9821), precision (0.9873 vs. 0.9810), recall (0.9889 vs. 0.9845), F1 score (0.9881 vs. 0.9827), specificity (0.9937 vs. 0.9912), and ROC AUC (0.9995 vs. 0.9991), as stated earlier in the result section of this work. We successfully implemented a stabilizing algorithm for the fuzzy membership boundary of pneumonia symptoms/conditions, and the DL based confidence score succeeded in reporting borderline cases, to avoid ambiguity in diagnosis. Thus, our work has immense potential as a robust and interpretable tool that can aid clinicians, especially in settings where expert judgment may be limited or variable.

While the proposed hybrid system shows promising diagnostic performance and interpretability, several limitations must be recognized. First, the dataset used for model training and evaluation lacks geographical and demographic details. This raises concerns about how our tool applies to diverse patient groups. Second, the current evaluation relies only on internal validation; external testing on independent, real-world clinical datasets is needed to confirm its reliability in various healthcare settings. Third, while the fuzzy inference system was developed with a clinician, it abstracts clinical decision logic, which may oversimplify complex relationships between symptoms. Finally, the system's severity adjustment depends on the DL-based confidence: While this helps refine predictions, it carries a risk of error propagation in cases of high-confidence misclassification. Future work will tackle these challenges through wider data integration, real-world pilot testing, and incorporating uncertainty-aware modelling strategies. Furthermore, the transformer-based models like ViT have been studied earlier in the case of pneumonia diagnosis, with effective results [61–63]. The inclusion of ViTs in future would enhance the model's ability to detect sequential symptom changes, as their integration with dynamic symptom monitoring or fuzzy inference systems remains rare.

In future, we plan to transit this work from experimental validation to clinical application as follows:

• DICOM Support: Incorporating DICOM (Digital Imaging and Communications in Medicine) standards will ensure compatibility with medical imaging workflows. It will also allow for direct ingestion of images from hospital radiology systems.

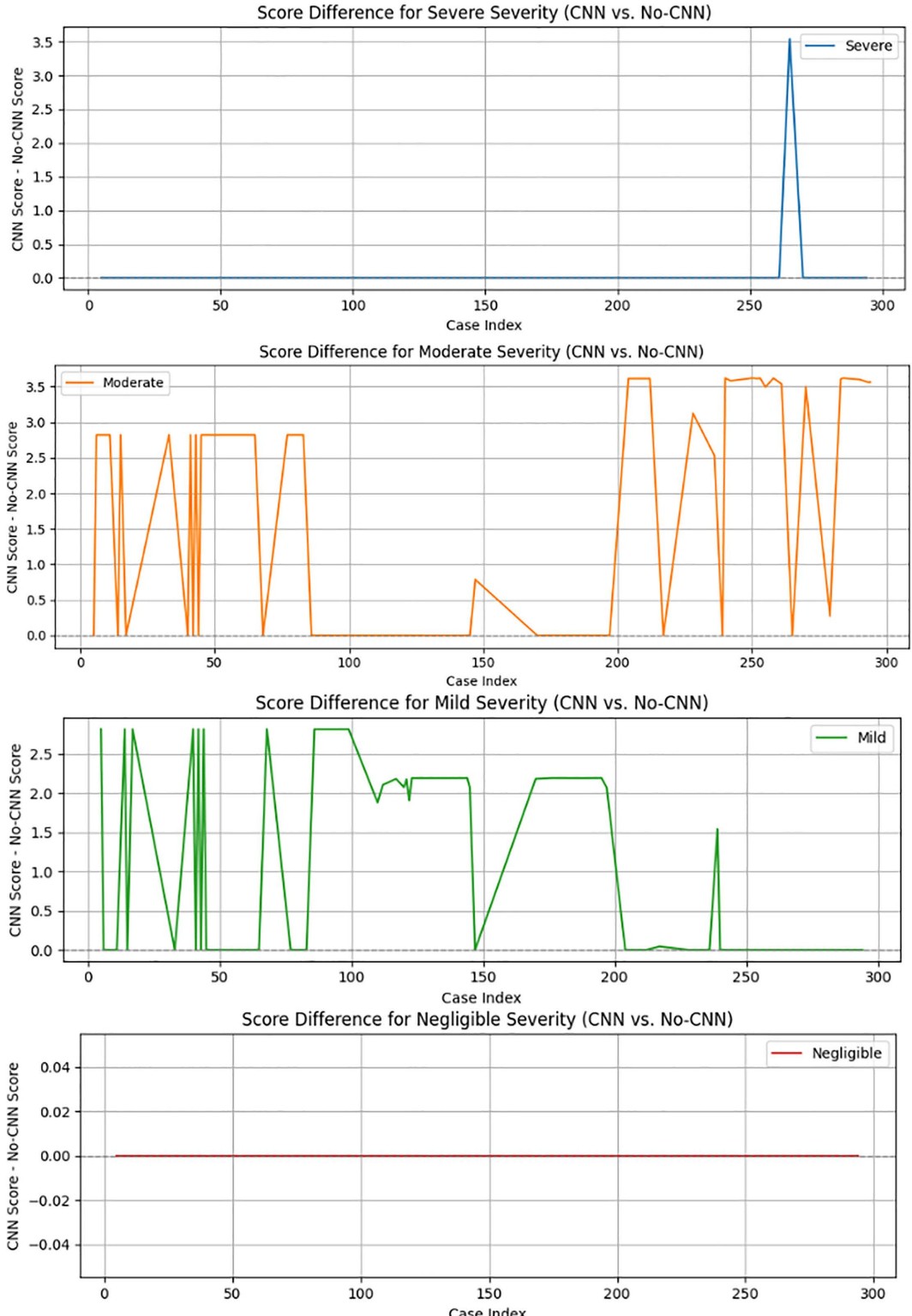

**Fig 12. Score difference across severities when CNN confidence (also referred to as DL confidence) is used vs. not used.** This figure compares the severity scores from the fuzzy system with and without CNN confidence across 300 test cases. Each subplot represents a specific severity class: Severe, Moderate, Mild, and Negligible. The y-axis shows the score difference (CNN – No CNN). The scores for Negligible and Severe are mostly

unchanged. This indicates that the CNN input had little effect in these cases. In contrast, Moderate and Mild show significant and frequent changes. This highlights that CNN confidence greatly impacts borderline or uncertain classifications. This supports the idea that CNN-guided fuzzy reasoning is especially useful in unclear situations.

- PACS APIs: By creating API-based connectors for Picture Archiving and Communication Systems (PACS), the model can work alongside radiologist tools. This allows for automatic triage and severity flagging within the current hospital setup.

To be more precise, we plan to carry out a pilot study with clinicians to evaluate how the model fits into actual clinical workflows. This study will gather feedback from radiologists and physicians on usability. It will also assess the impact on diagnostic time and accuracy and measure user trust. The insights gained will help refine the interface, improve interpretability, and ensure the system supports clinical decision-making. Our ultimate goal is to provide a reliable, explainable AI tool that improves pneumonia diagnosis, especially in resource-limited settings.

## Supporting information

**S1 Appendix.  This appendix includes insights from the collaborating physician regarding how fuzzy-based pneumonia symptom rules were developed, verified, and reasoned.** It contains S1 Appendix, which features the doctor's comments during the creation of these fuzzy-based symptoms and conditions for pneumonia.
(DOCX)

## Acknowledgments

We are grateful to Dr. Nishan Bhattrai for his valuable support in clinically validating the pneumonia symptoms used in the fuzzy logic framework. His details are as follows: Background: MBBS, MD (Internal Medicine), DM (Cardiology). Designation: Registrar, Department of Cardiology, Dhulikhel Hospital, Kavre, Nepal. E-mail: nishanbhattarai0213@gmail.com. Also, we would like to express our sincere gratitude to the Nepalese Education in E-health Masters (NEEM) project under GA101083048 at the Health Informatics Lab in Kathmandu University for supporting this research *(Sulav Baral and Rabindra Bista are co-first authors.)*

## Author contributions

**Conceptualization:** Rabindra Bista.

**Data curation:** Sulav Baral.

**Formal analysis:** Sulav Baral, Rabindra Bista.

**Investigation:** Sulav Baral, Rabindra Bista, Sanjog Sigdel.

**Methodology:** Rabindra Bista.

**Project administration:** Rabindra Bista.

**Resources:** Rabindra Bista, joao ferreira.

**Software:** Sulav Baral.

**Supervision:** Rabindra Bista.

**Validation:** Sulav Baral.

**Visualization:** Sulav Baral.

**Writing – original draft:** Sulav Baral.

**Writing – review & editing:** Rabindra Bista, joao ferreira.

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
