## [Decision Letter · Decision Letter 0]

22 May 2025

PONE-D-25-17872A Hybrid Dense Convolutional Network and Fuzzy Inference System for Pneumonia Diagnosis with Dynamic Symptom TrackingPLOS ONE

Dear Dr. ferreira,

Thank you for submitting your manuscript to PLOS ONE. After careful consideration, we feel that it has merit but does not fully meet PLOS ONE’s publication criteria as it currently stands. Therefore, we invite you to submit a revised version of the manuscript that addresses the points raised during the review process.

Please revise the paper according to the reviewer comments. Please submit your revised manuscript by Jul 06 2025 11:59PM. If you will need more time than this to complete your revisions, please reply to this message or contact the journal office at plosone@plos.org . Please include the following items when submitting your revised manuscript:

We look forward to receiving your revised manuscript.

Kind regards,

Fatih Uysal, Ph.D.

Academic Editor

PLOS ONE

Journal Requirements:

Additional Editor Comments :

Please revise the paper according to the reviewer comments.

Reviewers' comments:

Reviewer's Responses to Questions

**Comments to the Author**

1. Is the manuscript technically sound, and do the data support the conclusions?

Reviewer #1: Yes

Reviewer #2: Yes

2. Has the statistical analysis been performed appropriately and rigorously? 

Reviewer #1: Yes

Reviewer #2: N/A

3. Have the authors made all data underlying the findings in their manuscript fully available?

Reviewer #1: Yes

Reviewer #2: Yes

4. Is the manuscript presented in an intelligible fashion and written in standard English?

Reviewer #1: Yes

Reviewer #2: Yes

5. Review Comments to the Author

Reviewer #1: First, I would like to thank the author for this study. .This paper presents, A hybrid deep learning and fuzzy logic system is proposed for the diagnosis of pneumonia. The use of artificial intelligence in the field of medical imaging is an important issue, especially considering the lack of expert radiologists in low-resource settings, the potential of such systems is quite large.

When the paper is analyzed in detail, it is clear that serious revisions need to be made. The revision requests are listed as follows.

**The abstract does not explain Concepts such as ‘fuzzy membership degrees’ and ‘dynamic adjustment mechanism’. Reader-friendly language should be used, or the clinical importance of the terms should be emphasised in more detail.

**Demographic information (age, gender, geographical distribution), labelling process, and potential biases of the data set obtained from Kaggle are missing. It is appropriate to add these details for data quality and generalisability, which are essential for scientific studies.

**It is not explained how parameters such as ‘clip limit=2.0’ and ‘8x8 tile’ in CLAHE are determined. The effect of these choices on performance should be explained in detail.

**Why DenseNet201 was chosen over DenseNet121 and DenseNet169 is not analysed in depth. This decision should be explained in detail, along with its advantages.

**The superiority of the Capsule Network over traditional CNNs is not clear. It is important to support the contribution of this method with comparative results.

**It is unclear how the fuzzy rules are developed with clinical experts; it should be explained in detail.

**Confidence intervals or p-values of results, such as 97.96% accuracy, are missing. Statistical robustness of the results should be demonstrated.

**In the article, performance evaluation is based only on accuracy. This is inadequate, especially for medical diagnostic systems, so recall, F1-score, precision, specificity, and ROC AUC should be added to the article.

**The advantages/disadvantages of the Stratified K-Fold, Bootstrap, and Monte Carlo methods and why they were chosen are not discussed. Performance differences should be analysed in detail.

**How the system will be integrated in the hospital environment (e.g., compatibility with PACS) and clinician feedback are missing. A pilot study should be suggested.

Reviewer #2: I have reviewed the paper titled "A Hybrid Dense Convolutional Network and Fuzzy Inference System for Pneumonia Diagnosis with Dynamic Symptom Tracking." The study presents a hybrid AI-based approach combining DenseNet201 and a fuzzy inference system to classify pneumonia from chest X-ray images while adapting to symptom changes over time. The proposed integration of CNN confidence scores into the fuzzy system offers a level of interpretability and adaptability not commonly found in standard deep learning models.

Despite the strong performance metrics reported, the method section appears unnecessarily complex, and the novelty primarily lies in combining existing components rather than introducing a fundamentally new technique. The system’s dependence on CNN output for fuzzy logic adjustment could be more critically examined. External validation using real-world clinical data is missing, which limits the evidence for generalizability. Additionally, the fuzzy rules appear to be system-defined rather than clinician-informed, which raises concerns about medical relevance. The authors should explicitly state the study’s limitations and share their code and dataset to enhance transparency and reproducibility.

For methodological insight into pattern-based biomedical signal processing in another neurophysiological context, I recommend reviewing the paper “TPat: Transition pattern feature extraction based Parkinson’s disorder detection using FNIRS signals” (Tuncer et al., 2024), which proposes an explainable pattern extraction method relevant to clinical AI applications.

6. PLOS authors have the option to publish the peer review history of their article (what does this mean? ). If published, this will include your full peer review and any attached files.

**Do you want your identity to be public for this peer review?** For information about this choice, including consent withdrawal, please see our Privacy Policy .

Reviewer #1: No

Reviewer #2: No

---

## [Author Response · Author response to Decision Letter 1]

11 Jun 2025

Response to Reviewer #1:

Comment: The abstract does not explain Concepts such as ‘fuzzy membership degrees’ and ‘dynamic adjustment mechanism’. Reader-friendly language should be used, or the clinical importance of the terms should be emphasized in more detail.

Response: Thank you for the statement. We have modified the abstract, reducing the complexity of the language, to make it reader-friendly. We have maintained PLOS one’s 300-word limit for abstract as well.

Comment: Demographic information (age, gender, geographical distribution), labelling process, and potential biases of the data set obtained from Kaggle are missing. It is appropriate to add these details for data quality and generalizability, which are essential for scientific studies.

Response: We appreciate your response regarding our selection of the dataset. We have included all possible details of the Chest X-ray dataset in the “3.1 Data Collection” part of the 3. Methodology section. We also included the link to the dataset in the supplementary material. However, the dataset doesn’t have sufficient description regarding the demographic information, so we added this issue as our limitation.

Comment: It is not explained how parameters such as ‘clip limit=2.0’ and ‘8x8 tile’ in CLAHE are determined. The effect of these choices on performance should be explained in detail.

Response: With much pleasure! We have justified our selection of those parameters via sensitivity analysis in “3.3.1.1 Contrast Limited Adaptive Histogram Equalization (CLAHE)” under the methodology section.

Comment: Why DenseNet201 was chosen over DenseNet121 and DenseNet169 is not analyzed in depth. This decision should be explained in detail, along with its advantages.

Response: Thank you for this response! This is now highlighted in the Result section under sub subheading “4.3 Results of Cross-validation methods for pneumonia classification”. There we have included Table 8, which reports Training, Testing and Validation accuracy/loss for each model and cross-validation method used.

Comment: The superiority of the Capsule Network over traditional CNNs is not clear. It is important to support the contribution of this method with comparative results.

Response: Thank you very much for this feedback! We have included a paired t-test comparison demonstrating Capsule Network’s superiority over standard CNN with a Full connection, which is provided in “4.4 Significance of capsule layers” under the Results section of our paper.

Comment: It is unclear how the fuzzy rules are developed with clinical experts; it should be explained in detail.

Response: Acknowledged! We have described the Fuzzy system under the subheading “3.3 Machine learning approach” in the Methodology section. We have also mentioned the details of an expert cardiologist in the acknowledgement section, who helped us select the pneumonia symptoms during the development of the Fuzzy rules.

Comment: Confidence intervals or p-values of results, such as 97.96% accuracy, are missing. Statistical robustness of the results should be demonstrated.

Response: Due to the large number of columns needed for 4 separate cross validation methods, initially, we only included the training metrics such as Validation loss, Validation accuracy, Test loss and Test Accuracy in “4.3 Results of Cross-validation methods for pneumonia classification”. However, we have included statistical results in “4.4 Significance of capsule layers, where we again performed a paired t-test between Capsule Layer pipeline and a standard Full connection CNN.

To avoid confusion, 99.01% is the test accuracy achieved in the original stratified k-fold based training, and 98.76% is the test accuracy achieved by further using the bootstrap sampling method for the test data (aforementioned above about the paired t-test).

Comment: In the article, performance evaluation is based only on accuracy. This is inadequate, especially for medical diagnostic systems, so recall, F1-score, precision, specificity, and ROC AUC should be added to the article.

Response: Much obliged! We mentioned our preference for DenseNet201, so we have included per-class evaluation metrics across Stratified k-folds for the DenseNet201 model on the test dataset in “4.3 Results of Cross-validation methods for pneumonia classification” under the Results section. It includes Precision, Recall, F1-Score and Specificity for each of the 5 folds via a stratified k-fold validation cross-validation method.

Comment: The advantages/disadvantages of the Stratified k-Fold, Bootstrap, and Monte Carlo methods and why they were chosen are not discussed. Performance differences should be analyzed in detail.

Response: Thank you very much for pointing this out! We have included the advantages/strengths of each method in the “3.2 Splitting the dataset using cross-validation methods” under the methodology section. Subsection “4.3 Results of Cross-validation methods for pneumonia classification” under the Results section highlights the key metrics related to the performance achieved with different DenseNet models, which indicates our selection choice.

Comment: How the system will be integrated in the hospital environment (e.g., compatibility with PACS) and clinician feedback are missing. A pilot study should be suggested.

Response: Yes, very interesting question! We have reported the future scope and direction of our work in the Conclusion and future work section of this paper.

Response to Reviewer #2:

Comment: The method section appears unnecessarily complex.

Response: We appreciate this observation. To improve clarity, we have revised the Methods section to streamline explanations, reduce redundancy, and more clearly distinguish the roles of each component (CNN, fuzzy inference system, and dynamic adjustment mechanism). At the beginning of the Methodology section, we have included a flow diagram summarizing the architecture to support the reader's understanding.

Comment: The novelty primarily lies in combining existing components rather than introducing a fundamentally new technique.

Response: The novelty primarily lies in combining existing components rather than introducing a fundamentally new technique.

Comment: We agree that some Machine Learning components in our work have already been used in previous research as well, but we emphasize that the contribution lies in their integration and clinical framing. Our system uniquely combines a state-of-the-art DenseNet201 classifier combined with Capsule Network. The latter preserves important spatial features extracted from the Deep learning procedure. Also, we developed a fuzzy inference system by noting down appropriate pneumonia-related symptoms/conditions with help from an expert cardiologist. Moreover, our work combines the confidence score generated from the DenseNet classification into the fuzzy system as a membership value. By combining it with the unique Dynamic membership adjustment algorithm, we have identified borderline cases (a case compared between the use of Deep learning-based confidence and no Deep learning-based confidence, where only one output is close to a severity level) for 4 different severity stages of pneumonia (negligible, mild, moderate and severe). This ensures clinicians effectively identify pneumonia severity stage.

Comment: The system’s dependence on CNN output for fuzzy logic adjustment could be more critically examined.

Response: Thank you for this important point! We have included a sub section: 4.5 Influence of Confidence score on pneumonia membership range, under the Results section, which highlights the difference between using CNN confidence score (also referred to as Deep learning confidence score), and without the use of such confidence score for the fuzzy inference engine. We also explained why the use of a confidence metric is essential for a proper diagnosis. However, we also include a discussion on potential limitations of this dependency in the Conclusion and future work, stating that this work of ours hasn’t been deployed in real real-world scenario yet.

Comment: The fuzzy rules appear to be system-defined rather than clinician-informed.

Response: We thank the reviewer for this concern. In the subsection 3.3.5 under Methodology, we have further made a subsection 3.3.5.1, where we reported the appropriate pneumonia guidelines used to develop the fuzzy inference system. Furthermore, this part was developed in coordination with an expert cardiologist who has been acknowledged in the respective section of our work.

Comment: The authors should explicitly state the study’s limitations and share their code and dataset to enhance transparency.

Response: We agree. We have clearly defined the limitations of our work in the section Conclusion and future work. Furthermore, we have included a proper link to the Chest X-ray dataset and the GitHub link to the relevant materials of our work in the Supplementary material section. The repository will be made public once the paper becomes fully acceptable for this journal.

Comment: Comment: The reviewer recommends the paper “TPat” for insights into explainable biomedical AI.

Response: Thank you for the suggestion. We have properly cited the paper, and explained its importance in the subsection 2.5 under the Previous Study section.

---

## [Decision Letter · Decision Letter 1]

3 Jul 2025

PONE-D-25-17872R1A Hybrid Dense Convolutional Network and Fuzzy Inference System for Pneumonia Diagnosis with Dynamic Symptom TrackingPLOS ONE

Dear Dr. ferreira,

Thank you for submitting your manuscript to PLOS ONE. After careful consideration, we feel that it has merit but does not fully meet PLOS ONE’s publication criteria as it currently stands. Therefore, we invite you to submit a revised version of the manuscript that addresses the points raised during the review process. **Revise the paper based on referee comments.** Please submit your revised manuscript by Aug 17 2025 11:59PM. If you will need more time than this to complete your revisions, please reply to this message or contact the journal office at plosone@plos.org . Please include the following items when submitting your revised manuscript:

We look forward to receiving your revised manuscript.

Kind regards,

Fatih Uysal, Ph.D.

Academic Editor

PLOS ONE

**Journal Requirements:**

**Additional Editor Comments:**

Revise the paper based on referee comments.

Reviewers' comments:

Reviewer's Responses to Questions

**Comments to the Author**

1. If the authors have adequately addressed your comments raised in a previous round of review and you feel that this manuscript is now acceptable for publication, you may indicate that here to bypass the “Comments to the Author” section, enter your conflict of interest statement in the “Confidential to Editor” section, and submit your "Accept" recommendation.

Reviewer #1: All comments have been addressed

Reviewer #2: All comments have been addressed

2. Is the manuscript technically sound, and do the data support the conclusions?

Reviewer #1: Yes

Reviewer #2: Yes

3. Has the statistical analysis been performed appropriately and rigorously? 

Reviewer #1: Yes

Reviewer #2: Yes

4. Have the authors made all data underlying the findings in their manuscript fully available?

Reviewer #1: Yes

Reviewer #2: Yes

5. Is the manuscript presented in an intelligible fashion and written in standard English?

Reviewer #1: Yes

Reviewer #2: Yes

6. Review Comments to the Author

**Reviewer #1: ** all necessary revisions are sufficient and have been made. A few minor revisions are needed.

** I believe that comparing your work with the latest Pneumonia article will enrich your work. https://doi.org/10.3390/app15126487

**Reviewer #2: ** The authors have completely addressed all my comments, and I have no further concerns. Therefore, I recommend accepting the paper.

7. PLOS authors have the option to publish the peer review history of their article (what does this mean? ). If published, this will include your full peer review and any attached files.

**Do you want your identity to be public for this peer review?** For information about this choice, including consent withdrawal, please see our Privacy Policy .

Reviewer #1: No

Reviewer #2: No

---

## [Author Response · Author response to Decision Letter 2]

8 Sep 2025

Response to Reviewer #1:

Comment #1: All necessary revisions are sufficient and have been made. A few minor revisions are needed.

** I believe that comparing your work with the latest Pneumonia article will enrich your work. https://doi.org/10.3390/app15126487

Response: Thank you very much for your feedback! Under the Results section of this work, we included subtopic “4.3.1 Comparison with PELM” (Page 25), which compares the machine learning metrics achieved in that Pneumonia research with our research.

Apart from the comparison, we modified the Abstract section to increase clarity (to suit PLOS one’s readability standard), and made a few grammatical as well as factual corrections (Equation number 36 on page 30).

Response to Reviewer #2:

Comment: The authors have completely addressed all my comments, and I have no further concerns. Therefore, I recommend accepting the paper.

Response: Much obliged! Your feedback helped strengthen the scientific aspects of our paper.

---

## [Editor Report · Decision Letter 2]

6 Oct 2025

A Hybrid Dense Convolutional Network and Fuzzy Inference System for Pneumonia Diagnosis with Dynamic Symptom Tracking

PONE-D-25-17872R2

Dear Dr. ferreira,

We’re pleased to inform you that your manuscript has been judged scientifically suitable for publication and will be formally accepted for publication once it meets all outstanding technical requirements.

Kind regards,

Fatih Uysal, Ph.D.

Academic Editor

PLOS ONE

Additional Editor Comments (optional):

Considering the current status of the paper, it has been decided to accept it.
---

## [Editor Report · Acceptance letter]

PONE-D-25-17872R2

PLOS ONE

Dear Dr. ferreira,

I'm pleased to inform you that your manuscript has been deemed suitable for publication in PLOS ONE. Congratulations! Your manuscript is now being handed over to our production team.

Kind regards,

on behalf of

Dr. Fatih Uysal

Academic Editor

PLOS ONE